# Controlled disagreement improves generalization in decentralized training

## Abstract

Decentralized training is often regarded as inferior to centralized training because the consensus errors between workers are thought to undermine convergence and generalization, even with homogeneous data distributions. This work challenges this view by introducing decentralized SGD with Adaptive Consensus (DSGD-AC), which intentionally preserves non-vanishing consensus errors through a time-dependent scaling mechanism. We prove that these errors are not random noise but systematically align with the dominant Hessian subspace, acting as structured perturbations that guide optimization toward flatter minima. Across image classification and machine translation benchmarks, DSGD-AC consistently surpasses both standard DSGD and centralized SGD in test accuracy and solution flatness. Together, these results establish consensus errors as a useful implicit regularizer and open a new perspective on the design of decentralized learning algorithms.

## 1 Introduction

In large-scale deep learning, decentralized optimization, where workers exchange model parameters only with neighbors, reduces the overhead of global synchronization and avoids costly all-reduce communication (Abadi et al., 2016; Li et al., 2020). This neighbor-only exchange can substantially reduce communication overhead, latency, and single points of failure, making decentralized approaches attractive for geographically distributed systems (Dhasade et al., 2023; Gholami & Seferoglu, 2024) and even GPU clusters (Lian et al., 2017; Assran et al., 2019; Wang et al., 2025).

Despite its practical appeal in runtime efficiency, decentralized training methods such as Decentralized Stochastic Gradient Descent (DSGD) are conventionally viewed as suboptimal compared to centralized/synchronous SGD in terms of convergence and, importantly, generalization performance even with i.i.d. data distributions among workers. This gap is largely attributed to consensus errors — persistent discrepancies in the model parameters maintained by different workers (Alghunaim & Yuan, 2022; Zhu et al., 2022). Prior work has focused heavily on minimizing these consensus errors to close the gap. Massive efforts have been made to reduce consensus errors, which involve communication topologies (Ying et al., 2021; Takezawa et al., 2023) and algorithm designs (Pu & Nedić, 2021; Wang et al., 2019; Lin et al., 2021) to ensure decentralized training closely approximates centralized training.

However, the conventional perspective neglects the potential constructive role of consensus errors. Rather than detrimental noise, these discrepancies may serve as structured perturbations that facilitate exploration of flatter minima in the loss landscape — solutions known to correlate with superior generalization (Jiang et al., 2019). This insight draws inspiration from sharpness-aware minimization strategies (Foret et al., 2020; Bisla et al., 2022; Li et al., 2024b), which explicitly introduce curvature-aware perturbations to enhance model robustness and performance.

In this study, we challenge the conventional view by introducing Decentralized SGD with Adaptive Consensus (DSGD-AC), an algorithm that strategically preserves non-vanishing consensus errors through an adaptive, time-dependent scaling mechanism. We provide a theoretical analysis demonstrating that consensus errors align with the dominant subspace of the Hessian matrix, thereby inducing beneficial curvature-related perturbations from the global average. Notably, DSGD-AC incurs negligible additional computational overhead relative to standard SGD or DSGD and enjoys the superior runtime efficiency over SGD at the same time.

Comprehensive experiments reveal that DSGD-AC consistently surpasses both DSGD and centralized SGD in terms of test accuracy and the flatness of the attained minima. To the best of our knowledge, this work constitutes the first demonstration of decentralized training outperforming centralized approaches under optimal conditions by a clear margin.

The main contributions of this work are: (1) the proposal of DSGD-AC, an adaptive consensus algorithm that maintains theoretically-justified non-vanishing consensus errors at minimal computational expense, (2) a theoretical characterization of consensus error and its alignment with the dominant Hessian subspace, and (3) empirical validation of the superior generalization performance of DSGD-AC on typical deep learning tasks.

## 1.1 RELATED WORK

**Canonical view about consensus errors**  The prevailing perspective on decentralized training is that it should approximate synchronous/centralized training as closely as possible. To mitigate discrepancies among local models caused by weakly connected networks, prior work has focused on tracking global information (Wang et al., 2019; Pu & Nedić, 2021; Yuan et al., 2021; Takezawa et al., 2022), enhancing communication topologies to improve convergence rates (Ying et al., 2021; Zhu et al., 2022; Takezawa et al., 2023), and more. In addition, several theoretical studies (Zhu et al., 2022; Alghunaim & Yuan, 2022) establish a theoretical connection between the connectivity of decentralized communication topologies and both convergence and generalization, demonstrating that weaker connectivity results in poorer outcomes on both fronts. In contrast, we demonstrate the potential advantages of the consensus error by identifying its correlation with the dominant Hessian subspace, and we propose DSGD-AC in which consensus errors can, in practice, outperform SGD in deep learning tasks.

**Explorations beyond the canonical view**  As the canonical perspective dominates, the effort that has been made towards suggesting potential benefits of consensus errors is limited. Kong et al. (2021) conducts empirical studies aimed at identifying thresholds of consensus errors. Although they highlight advantages of consensus errors in certain phases, the regime where consensus errors exceed those of DSGD with a ring topology remains unexplored, and the consensus control scheme proposed in the work does not yield clear performance improvements. Zhu et al. (2023) offers a novel interpretation, framing consensus errors in DSGD as random perturbations around the global average, which are asymptotically equivalent to average-direction SAM (Bisla et al., 2022). Our work further identifies the intrinsic curvature-related property of the consensus errors, and, by proposing DSGD-AC, empirically demonstrates the superior potential of decentralized training over centralized training without being limited to the large-batch setting.

**Explicit curvature-related perturbations improve generalization but are costly**  With the idea of taking the global average as the deployed model (Zhu et al., 2023), decentralized learning can be interpreted as the learning on the (virtual) global average with the workers as the perturbed global average. Sharpness-aware minimization (SAM) was first proposed by Foret et al. (2020) to improve the generalization of deep neural networks, and many variants (Kwon et al., 2021; Bisla et al., 2022; Liu et al., 2022; Li et al., 2024a; Luo et al., 2024) were developed to further improve SAM. However, to achieve the best performance, the algorithms typically require one or more additional rounds of gradient evaluations, which significantly increase the computational costs. Our work utilizes the potential of the consensus errors as free perturbations to enhance the generalization without introducing extra computation.

## 2 PROBLEM SETTINGS AND NOTATIONS

**Practical remarks on data distributions and the distributed data sampler**  Our work focuses on decentralized training in GPU clusters where the whole dataset can be easily accessed by all workers. This scenario is also widely studied in many other literature (Assran et al., 2019; Ying et al., 2021; Kong et al., 2021; Wang et al., 2025), and is important for improving the efficiency of large-scale distributed training. The common practice for the distributed data sampler (also used in our experiments) is to reshuffle the full dataset at the start of each epoch and partition it evenly across workers. The strategy ensures, in expectation, i.i.d. data distributions among workers.

**Decentralized Optimization**   We denote the set of integers $\{1, 2, \cdots, k\}$ by $[k]$, the number of workers by $n \in \mathbb{N}^+$, and the dimension of model parameters by $d \in \mathbb{N}^+$. In the standard decentralized optimization setup with $n$ workers, each worker $i \in [n]$ holds a local objective determined by its local dataset $\mathcal{D}_i$:

$$f_i(x) = \mathbb{E}_{s \sim \mathcal{D}_i}[f_i(x; s)], \tag{1}$$

and the optimization problem is the local losses evaluated on the local models with a hard consensus constraint:

$$\underset{\{x_1, x_2, \cdots, x_n\}}{\text{minimize}} \; F(x_1, \cdots, x_n) = \frac{1}{n} \sum_{i=1}^{n} f_i(x_i), \quad \text{subject to } x_i = x_j, \forall i, j \in [n] \tag{2}$$

In the i.i.d. data distribution setting, we have $f_i = f_j = F$ for all $i, j \in [n]$.

**Decentralized SGD (DSGD)**   The update of DSGD (Lian et al., 2017) on worker $i$ is:

$$x_i^{(t)} = x_i^{(t-1)} - \alpha^{(t)} \nabla f(x_i^{(t-1)}; s_i^{(t)}) + \sum_{j \in \mathcal{N}(i)} W_{ij}(x_j^{(t-1)} - x_i^{(t)}) \tag{3}$$

where $\mathcal{N}(i)$ is the set of neighbors of worker $i$ (including itself), $W$ is a symmetric, non-negative, and doubly stochastic matrix defining the weights of the edges ($W_{ij} = 0$ if worker $i$ is not a neighbor of worker $j$), and $s_i^{(t)}$ denotes the stochastic mini-batch sampled by worker $i$ at iteration $t$.

Following the common notations in decentralized optimization, we denote the global average by $\bar{x}^{(t)} := \sum_{i=1}^{n} x_i^{(t)}$, the consensus error of worker $i$ by $e_i^{(t)} := x_i^{(t)} - \bar{x}^{(t)}$, the matrix form of all local models by $X^{(t)} := [x_1^{(t)}, \cdots, x_n^{(t)}] \in \mathbb{R}^{d \times n}$, the matrix form of all local stochastic gradients by $G^{(t)} := [\nabla f_1(x_1^{(t-1)}; s_1^{(t)}), \cdots, \nabla f_n(x_n^{(t-1)}; s_n^{(t)})]$, the matrix form of all consensus errors as $\Delta^{(t)} = [e_1^{(t)}, \cdots, e_n^{(t)}]$, and the matrix $\bar{X}$ by $\bar{X}^{(t)} = [\bar{x}^{(t)}, \cdots, \bar{x}^{(t)}]$.

Note that there is another variant of DSGD that performs the local update before communication. We focus on the variant in Eq. (3) as it is shown to be more efficient (Lian et al., 2017; Wang et al., 2025), and two variants are proven to have the same generalization bound (Bellet et al., 2023).

# 3   DSGD-AC: DECENTRALIZED SGD WITH ADAPTIVE CONSENSUS

In this section, we use the experiment of training a wide ResNet (WRN28-10) (Zagoruyko & Komodakis, 2016) on CIFAR-10 (Krizhevsky et al., 2009) as a showcase to demonstrate the limitation of DSGD and the improvement brought by our proposed algorithm. In the experiment, we employ the standard cosine annealing learning rate schedule (Loshchilov & Hutter, 2016) with a linear warm-up during the first 10 epochs (Figure 1, left). This learning rate schedule is commonly used in practice and can strike a balance between the training stability and generalization (Gotmare et al., 2018; Kalra & Barkeshli, 2024). For decentralized training, we use 8 workers and the one-peer ring as the decentralized communication topology.

## 3.1   FINDING: VANISHING CONSENSUS ERROR IN DSGD

We start by empirically investigating the dynamics of consensus errors when trained with DSGD. We track the average norm of the consensus errors during the training. We observe that, for DSGD, the consensus errors gradually vanish as the learning rate decreases (Figure 1, right).

From the theoretical perspective, by interpreting the mixing step as a gradient step on a quadratic consensus penalty, one obtains the per-step surrogate

$$
\begin{aligned}
&J^{(t)}(x_1, \cdots, x_n) \\
&= \sum_{i=1}^{n} f_i(x_i^{(t)}) + \frac{1}{2\alpha^{(t)}} \sum_{i,j \in [n]} W_{ij} \|x_i^{(t)} - x_j^{(t)}\|^2 \\
&= \underbrace{\sum_{i=1}^{n} f_i(\bar{x}^{(t)})}_{\text{objective on deployed model}} + \underbrace{\sum_{i=1}^{n} [f_i(x_i^{(t)}) - f_i(\bar{x}^{(t)})]}_{\text{sharpness}} + \underbrace{\frac{1}{2\alpha^{(t)}} \sum_{i,j \in [n]} W_{ij} \|x_i^{(t)} - x_j^{(t)}\|^2}_{\text{consensus regularizer}}
\end{aligned}
\tag{4}
$$

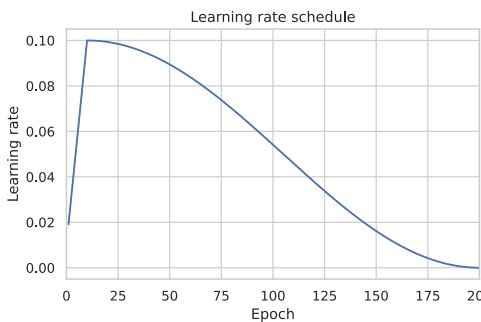 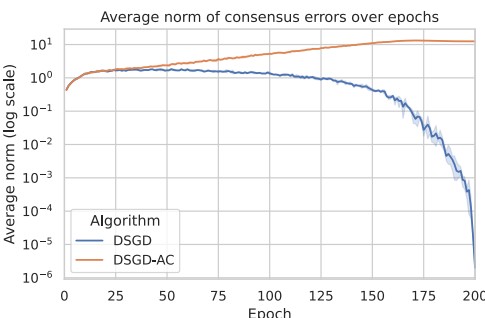

Figure 1: Decentralized training of WRN28-10 on CIFAR-10 (3 random runs for each algorithm) with 8 workers, and the communication topology is the one-peer ring topology. **Left**: Learning rate schedule (same for both algorithms). **Right**: Average norm of consensus errors evaluated at the end of every epoch ($\frac{1}{N}\sum_{i=1}^{N}\|x_i^{(eT)} - \bar{x}^{(eT)}\|$). $p$ is set to 3 for DSGD-AC.

With symmetric mixing weights and no momentum or adaptivity, each DSGD step is exactly a (stochastic) gradient on $J$. Thus, when $\alpha^{(t)}$ goes to 0, the consensus regularizer dominates the objective function, which minimizes the consensus errors. If considering this surrogate function, the empirical observation is not surprising because it reflects the hard constraint in the optimization problem in Eq. (2). However, the vanishing consensus errors void the sharpness term in Eq. (4) because the sharpness term because $f_i(x_i^{(t)}) \approx f_i(\bar{x}^{(t)})$ as $x_i^{(t)} - \bar{x} \to 0$. The only term left that is relevant to the deployed model $\bar{x}^{(t)}$ is the first term, which is the same objective as in synchronous SGD. Therefore, to maintain the potential benefits of free sharpness-aware regularization (Zhu et al., 2023) by the consensus errors, we need to maintain a non-vanishing radius throughout the training.

### 3.2 Algorithm: Decentralized SGD with adaptive consensus

The proposed algorithm is shown in Algorithm 1. The difference from DSGD is highlighted, and, compared with DSGD, the proposed variant includes an adaptive factor to maintain non-diminishing consensus errors intentionally. At the end of training, the algorithm takes the global average of all local models as the deployed model.

---

**Algorithm 1:** Decentralized SGD with adaptive consensus (DSGD-AC) on worker $i$

**Data:** Dataset ($D$), the number of workers ($N$), the number of epoch ($E$), the number of
    batches per epoch ($T$), intialization ($x^{(0)}$), and a hyperparameter ($p \in \mathbb{R}^+$) .

**Result:** Deployed model $\bar{x} = \frac{1}{n}\sum_{j=1}^{n} x_j^{(TE)}$

$x_1^{(0)} = x_2^{(0)} = \cdots = x_n^{(0)} = x^{(0)}$
**for** $t = 1$ *to* $TE$ **do**
$\quad g_i^{(t)} = \nabla f(x_i^{(t-1)}; s_i^{(t)})$
$\quad \gamma^{(t)} = \left[\alpha^{(t)}/\alpha_{\max}\right]^p$
$\quad x_i^{(t)} = x_i^{(t-1)} - \alpha^{(t)} g_i^{(t)} + \gamma^{(t)} \sum_{j \in \mathcal{N}(i)} W_{ij}(x_j^{(t-1)} - x_i^{(t-1)})$
**end**

---

Note that $\alpha^{(t)}$ is determined by the learning rate scheduler like cosine annealing (Loshchilov & Hutter, 2016), and $\alpha_{\max}$ is the maximal learning rate throughout the training, which ensures $\gamma^{(t)}$ is in the range $[0, 1]$.

We evaluate the performance of DSGD-AC on classical deep learning tasks in Section 4. In the numerical experiments, the results demonstrate the superior generalization performance of DSGD-

AC over DSGD and centralized SGD. We will analyze the reasons behind this by showing that DSGD-AC maintains non-diminishing and useful consensus errors in the following sections.

### 3.3 CONTROLLED CONSENSUS ERRORS IN DSGD-AC

The motivation of DSGD-AC is to maintain non-diminishing consensus errors. Therefore, we multiply the weight of the consensus regularizer in Eq. (4) by an adaptive $\gamma$, which directly leads to the DSGD-AC algorithm. The per-step surrogate function of DSGD-AC is mostly the same as that of DSGD. Only the weight of the consensus regularizer becomes $\gamma^{(t)}/(2N\alpha^{(t)})$.

In this section, we investigate the impact of $p$ on the magnitude of consensus errors. First, we can rewrite the update of DSGD-AC in matrix form,

$$X^{(t)} = X^{(t-1)} - \alpha^{(t)}G^{(t)} - \gamma^{(t)}X^{(t-1)}(I - W) = X^{(t-1)}(I - \gamma^{(t)}L) - \alpha^{(t)}G^{(t)} \quad (5)$$

where we denote the Laplacian matrix $L$ by $L = I - W$.

By subtracting $\bar{X}^{(t)}$ on both sides of Eq. (5) and using the fact that $\Delta^{(t)} = X^{(t)}(I - \frac{1}{n}\mathbf{1}\mathbf{1}^\top)$, the dynamics of consensus errors $\Delta^{(t)}$ can be derived as

$$
\begin{aligned}
\Delta^{(t)} &= X^{(t-1)}(I - \gamma^{(t)}L) - \alpha^{(t)}G^{(t)} - \bar{X}^{(t)} \\
&= X^{(t-1)}(I - \gamma^{(t)}L) - \alpha^{(t)}G^{(t)} - \bar{X}^{(t-1)} + \alpha^{(t)}G^{(t)} \cdot \frac{1}{n}\mathbf{1}\mathbf{1}^\top \\
&= \Delta^{(t-1)}(I - \gamma^{(t)}L) - \alpha^{(t)}G^{(t)}(I - \frac{1}{n}\mathbf{1}\mathbf{1}^\top)
\end{aligned}
\quad (6)
$$

Next, we denote $P = I - \frac{1}{n}\mathbf{1}\mathbf{1}^\top$, perform an eigen-decomposition of $L = U_L\Lambda_L U_L^\top$, and multiply Eq. (6) by $U_L$ from the right to obtain

$$
\begin{aligned}
\Delta^{(t)}U_L &= \Delta^{(t-1)}(I - \gamma^{(t)}U_L\Lambda_L U_L^\top)U_L - \alpha^{(t)}G^{(t)}P_L U_L \\
&= \Delta^{(t-1)}U_L(I - \gamma^{(t)}\Lambda_L) - \alpha^{(t)}G^{(t)}PU_L
\end{aligned}
\quad (7)
$$

By introducing $Z^{(t)} = \Delta^{(t)}U_L$ and $\tilde{G}^{(t)} = G^{(t)}PU_L$, we can re-write the update as

$$Z^{(t)} = Z^{(t-1)}(I - \gamma^{(t)}\Lambda_L) - \alpha^{(t)}\tilde{G}^{(t-1)} \quad (8)$$

Here, $Z^{(t)}$ collects the consensus error expressed in the eigenbasis of the Laplacian. The $k$-th row $Z_k^{(t)}$ contains the coefficients of $\Delta^{(t)}$ along the $k$-th Laplacian eigenvector, or *network mode*, and thus describes a characteristic pattern of disagreement across agents induced by the communication graph. We measure the overall amount of disagreement by the *disagreement radius*

$$r_t^2 := \mathbb{E}\left[\|\Delta^{(t)}\|_F^2\right].$$

Since $U_L$ is orthogonal, $\|\Delta^{(t)}\|_F = \|Z^{(t)}\|_F$, so the radius can be equivalently studied through the Laplacian-mode dynamics of $Z^{(t)}$ in Eq. (8). By analyzing these dynamics, we obtain the following proposition; the proof is deferred to Section A.1 in the appendix.

**Proposition 1 (Disagreement radius and the role of $\gamma$)** *In a quasi-stationary regime with mild bounded-moment and spectral assumptions (see Appendix A.1) the disagreement radius satisfies*

$$r_t^2 = \Theta\left(\frac{(\alpha^{(t)})^2}{\gamma^{(t)}}\right)$$

*In particular, if $\alpha^{(t)} \to 0$ and $\gamma^{(t)}$ is bounded away from zero, then $r_t^2 \to 0$. Thus, no constant $\gamma^{(t)}$ can maintain a non-vanishing disagreement radius under diminishing stepsizes. However, if we choose $\gamma^{(t)} = g_0 (\alpha^{(t)})^p$ for some $g_0 > 0$ and $p > 0$, then as $\alpha^{(t)} \to 0$,*

$$p < 2 \Rightarrow r_t^2 \to 0, \qquad p = 2 \Rightarrow r_t^2 = \Theta(1), \qquad p > 2 \Rightarrow r_t^2 \to \infty.$$

*and $p \geq 2$ is necessary to keep the consensus errors at a nontrivial scale.*

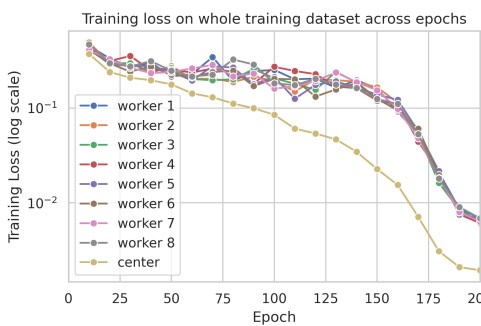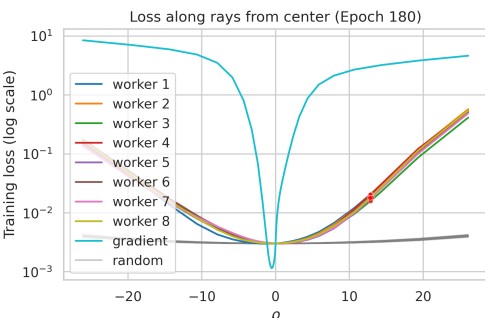

Figure 2: **Left**: Losses on the whole training dataset at local workers and global average. The losses are evaluated every 10 epochs. **Right**: Training loss at epoch 180 along: (1) `worker i`: lines connecting global average and worker $i$, (2) `gradient`: the line that aligns with the full-batch gradient at the global average and crosses the global average, and (3) `random`: 500 lines that cross the global average and follow random directions generated as in (Bisla et al., 2022). The $x$-axis means the directional magnitude of the perturbation along these directions. The red dots represent the losses at the local models. The losses are computed on $\sim 1/4$ of the training dataset due to computation complexity.

The proposition establishes that DSGD-AC maintains a nontrivial level of consensus errors throughout iterations. In fact, the proof of the proposition shows that the effective disagreement radius $r_t^2 = \mathbb{E}[\|\Delta^{(t)}\|_F^2]$ is on the order of $(\alpha^{(t)})^2/\gamma^{(t)}$. Since it has been empirically observed (see, *e.g.*, Bisla et al. (2022); Li et al. (2024a)) that it is advantageous to increase the radius slightly towards the end of the training, we chose $\gamma^{(t)} = g_0(\alpha^{(t)})^p$ with $p = 3$. Under cosine learning rate schedules, this choice induces a mild uptick in the radius toward the final stages of training, as illustrated in Figure 1 (right). A detailed sensitivity analysis in Appendix A.5.4 further supports the theory, demonstrating radius shrinkage for $p < 2$ and growth for $p > 2$ as $\alpha^{(t)} \to 0$ (see Figure 13).

### 3.4 CONSENSUS ERRORS ALIGN WITH DOMINANT SUBSPACE OF HESSIAN

Even though DSGD-AC maintains non-vanishing consensus errors, its role in leading to flatter minima and better generalization remains underexplored. In (Zhu et al., 2023, Theorem 1), consensus errors are interpreted as random perturbations within the subspace defined by the weight diversity matrix, and the resulting (asymptotically equivalent) average-direction SAM effect is shown to improve generalization. While this connection is insightful, the intrinsic structure of consensus errors (or the weight diversity matrix) has not been examined in detail.

To study this structure, we compare the training losses at local models with those at their global average. As shown in Figure 2 (left), the global average consistently achieves lower training loss than any individual worker. To further distinguish consensus errors from random perturbations, we evaluate the training losses along the directions of consensus errors and compare them against losses along sufficiently many random directions. Figure 2 (right) shows that the random directions are almost flat, which is expected given the large parameter space ($\sim$36M in WRN28-10) and the low-rank structure of the Hessian (Gur-Ari et al., 2018; Song et al., 2024). It is also consistent with empirical observations in (Keskar et al., 2016).

In contrast, directions induced by consensus errors yield significant increases in training loss, highlighting that these errors are not random but aligned with directions of higher curvature. This phenomenon suggests a correlation between consensus errors and the dominant subspace of the Hessian (or directions with larger curvature). Motivated by this observation, we formalize it in the following proposition, with the proof deferred to Appendix A.2.

**Proposition 2 (Structure of consensus errors)** *Let $x^*$ be a locally strongly convex minimizer of $F$ with Hessian $H = U_H \Lambda_H U_H^\top$ and eigenvalues $0 < \lambda_1(H) \leq \cdots \leq \lambda_d(H)$. Assume i.i.d. local objectives ($f_i \equiv F$), and let $W$ be a symmetric, doubly stochastic communication matrix associated with a connected undirected graph. Let $\Delta^{(t)}$ denote the consensus error at time $t$. Consider the*

*DSGD-AC recursion in a neighbourhood of $x^*$ and approximate the local gradients by their first-order Taylor expansion, $\nabla f_i(x_i^{(t)}) \approx H(x_i^{(t)} - x^*)$. For non-increasing stepsizes $\alpha^{(t)} > 0$ and consensus factors $\gamma^{(t)} > 0$, the projection $\Delta_k^{(t)} := u_k^\top \Delta^{(t)}$ of the consensus error onto each Hessian eigenvector $u_k$ then evolves as a scalar linear system whose stability requires*

$$\alpha^{(t)} < \frac{2 + (\lambda_{\min}(W) - 1)\,\gamma^{(t)}}{\lambda_k(H)}, \tag{9}$$

*where $\lambda_{\min}(W)$ is the smallest eigenvalue of $W$. The right-hand side of Eq. (9) is decreasing in $\lambda_k(H)$, so with non-increasing stepsizes $\alpha^{(t)}$ modes corresponding to smaller eigenvalues enter the stable regime earlier, while high-curvature modes remain closer to instability for longer and therefore retain higher variance under the same injected noise.*

**Remark 1 (Theoretical benefit of adaptive consensus)** A smaller consensus factor $\gamma^{(t)}$ relaxes the stability condition in (9). As $\gamma^{(t)}$ decreases during training, more low-curvature modes become stable, while the high-curvature modes remain closer to instability. As a result, the consensus errors gradually concentrate on a lower-dimensional subspace spanned by the dominant Hessian directions.

**Remark 2 (Alignment is meaningful only with a controlled disagreement radius)** The conclusion of Proposition 2 only holds when the disagreement radius stays in a reasonable range. Although one may relax the condition (9) by the selection of $W$ and $\gamma^{(t)}$, taking $\gamma^{(t)}$ too small or using a graph with a very large $\lambda_{\min}(W)$ can cause the disagreement to grow quickly (Proposition 1). In that case, the iterates may move out of the region where the local Taylor approximation is accurate. On the other hand, if the radius becomes too small, the disagreement barely perturbs the model, and its directional structure becomes unimportant. Thus, the alignment effect is useful only when the disagreement radius is neither too large nor too small.

To connect the result of Proposition 2 to the objective optimized by DSGD-AC, Appendix A.3 analyzes the deployed model $\bar{x}(t)$ and the disagreements $\delta_i^{(t)} = x_i^{(t)} - \bar{x}^{(t)}$. Using a second-order expansion of $F$ around $x^*$, we show that

$$\frac{1}{N} \sum_{i=1}^N f_i\big(x_i^{(t)}\big) = F(\bar{x}(t)) + \tfrac{1}{2}\operatorname{tr}(H\Sigma_t) + O\big((\operatorname{tr}\Sigma_t)^{3/2}\big),$$

where $\Sigma_t$ is the disagreement covariance. Thus, in this local regime, DSGD-AC can be interpreted as minimizing the central loss $F(\bar{x}(t))$ plus a Hessian-weighted disagreement penalty. A spectral decomposition of this penalty reveals that the mode weights are strictly increasing in the Hessian eigenvalues. Disagreement in sharper directions therefore incurs a larger penalty, resulting in a "curvature tilt" toward flatter minima; see Appendix A.3 for more details.

Given the alignment between the consensus errors and the dominant subspace, DSGD-AC can be interpreted as optimization over $\bar{x}$ with curvature-correlated noises, which has been both empirically and theoretically studied by many works (Foret et al., 2020; Zhang et al., 2023; Luo et al., 2024; Benedetti & Ventura, 2024). By maintaining non-vanishing consensus errors along with its regularization effect on the curvature of the loss landscape, DSGD-AC is expected to achieve better generalization performance than DSGD and SGD.

While the alignment exists and can be shown theoretically, the alignment is noisy and spans on less-sharp directions when compared with the gradient direction. As shown in Figure 2 (right), the gradient computed on the corresponding dataset leads to a sharper increase than the consensus errors. An interesting direction for future work could be an improved algorithm based on DSGD-AC that can utilize the gradient information to promote the concentration of consensus errors on the dominant Hessian subspace with small computational overhead.

## 4 NUMERICAL EXPERIMENTS

In this section, we present the results of the numerical experiments on image classification with wide ResNet (Zagoruyko & Komodakis, 2016) and on machine translation with transformers (Vaswani et al., 2017). In the experiments, we follow hyperparameters in the corresponding original papers, and we reproduce the same baseline performance for a fair comparison. For DSGD-AC, we use

$p = 3$ in all experiments. We defer the other hyperparameter details to Appendix A.4 and the sensitivity analysis on $p$ to Appendix A.5.4.

Each set of experiments consists of three random runs with fixed random seeds. We report $1\times$ standard deviation in all tables, and the shaded areas in plots correspond to the 95% confidence interval.

### 4.1 IMAGE CLASSIFICATION WITH WIDE RESNET

We train two variants of Wide ResNets (WRN28-10 and WRN16-8) (Zagoruyko & Komodakis, 2016) on two datasets, CIFAR-10 and CIFAR-100 (Krizhevsky et al., 2009). The classification accuracies on the test set and training/test losses of WRN28-10 on CIFAR-10 are shown in Figure 3. The test performance and the flatness of the solutions are reported in Table 1. The curves and statistics of the remaining experiments are deferred to Appendix A.5.1. Due to space limitations, we defer the results with varying topologies and worker counts to Appendix A.5.2 and A.5.3.

Since finding the best sharpness metric that always reflects the potential generalization is still an open question, we use the top-1 eigenvalue as a surrogate, which is widely used in other literature and proven to have a strong correlation (Bisla et al., 2022; Luo et al., 2024).

| Model | Dataset | Algorithm | Test Acc. (%) ↑ | Test Loss ↓ | Top-1 Eigenvalue ↓ |
|---|---|---|---|---|---|
| WRN28-10 | CIFAR-100 | DSGD | 79.86 ± 0.22 | 0.899 ± 0.008 | 49.57 ± 4.80 |
| | | SGD | 80.15 ± 0.42 | 0.878 ± 0.020 | 37.37 ± 2.88 |
| | | DSGD-AC | **82.38 ± 0.09** | **0.755 ± 0.008** | **19.80 ± 0.66** |
| | CIFAR-10 | DSGD | 96.07 ± 0.13 | 0.176 ± 0.005 | 22.43 ± 3.99 |
| | | SGD | 95.96 ± 0.14 | 0.182 ± 0.004 | 16.84 ± 0.32 |
| | | DSGD-AC | **96.77 ± 0.11** | **0.128 ± 0.003** | **8.96 ± 0.35** |
| WRN16-8 | CIFAR-100 | DSGD | 79.25 ± 0.26 | 0.854 ± 0.016 | 36.19 ± 3.80 |
| | | SGD | 79.42 ± 0.18 | 0.849 ± 0.015 | 33.77 ± 0.78 |
| | | DSGD-AC | **80.67 ± 0.11** | **0.771 ± 0.005** | **19.81 ± 0.16** |
| | CIFAR-10 | DSGD | 95.94 ± 0.11 | 0.152 ± 0.001 | 18.19 ± 0.64 |
| | | SGD | 95.81 ± 0.13 | 0.153 ± 0.003 | 17.49 ± 1.61 |
| | | DSGD-AC | **96.17 ± 0.04** | **0.129 ± 0.003** | **11.82 ± 0.48** |

Table 1: Performance comparison of DSGD, SGD, and DSGD-AC on image classification with wide ResNet with 8 workers. The top-1 eigenvalue is computed on the whole training set and approximated using the power iteration. The one-peer ring is used for decentralized training.

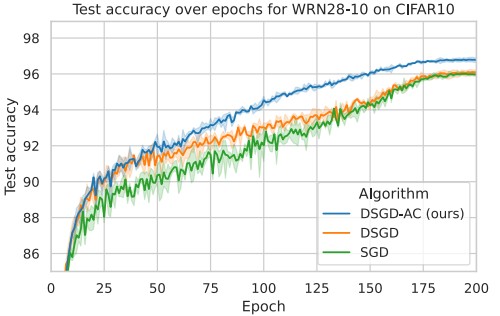
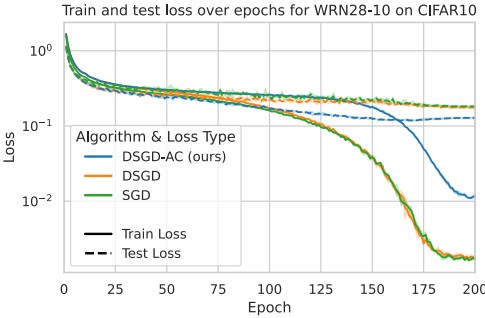

Figure 3: WRN28-10 on CIFAR-10. **Left**: Test accuracy on test set. For decentralized training, the accuracy is evaluated on the global average model. **Right**: Training losses (evaluated on the workers for decentralized training, and evaluated on perturbed points for SAM) and test losses (evaluated on the global average model for decentralized training). The curves for each algorithm are based on 3 runs with the same set of random seeds.

In the experiment results, DSGD-AC outperforms DSGD and SGD in test accuracy, test losses, and solution flatness by a clear margin. Moreover, it can also be seen that DSGD can not outperform SGD with its best performance.

## 4.2 MACHINE TRANSLATION WITH TRANSFORMERS

We also validate the idea of controlling consensus errors on transformer models by simply replacing the local update with the Adam optimizer (Adam et al., 2014). DSGD-AC is then adapted to DAdam-AC. We train Transformer (the big variant, ~213M parameters) (Vaswani et al., 2017) on WMT14 (English-to-German) (Bojar et al., 2014) and present the curves of training losses and BLEU scores on the test set. The BLEU scores (Papineni et al., 2002) (which is used to evaluate the translation quality in the original paper) and the losses on the test set and the training set are reported in Table 2.

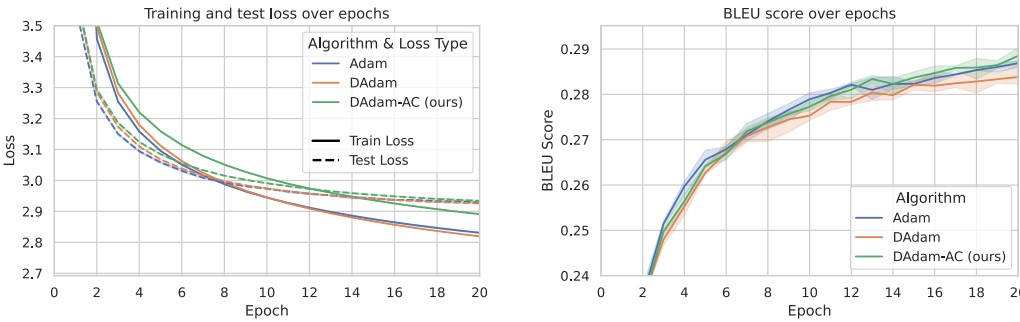

Figure 4: Transformer (big) on WMT14 English-to-German. **Left**: Losses on training set. **Right**: BLEU scores on the test set.

| Algorithm | BLEU score ↑ | Test loss ↓ | Train loss ↓ |
|-----------|--------------|-------------|--------------|
| Adam | 28.68 ± 0.07 | 2.9290 ± 0.0026 | 2.8310 ± 0.0019 |
| DAdam | 28.38 ± 0.22 | 2.9258 ± 0.0018 | **2.8195 ± 0.0008** |
| DAdam-AC | **28.89 ± 0.17** | **2.9205 ± 0.0020** | 2.8456 ± 0.0016 |

Table 2: Performance comparison of DAdam, Adam, and DAdam-AC on neural machine translation with the transformer model.

The results demonstrate that DAdam-AC can outperform other baselines on the translation quality metric and the test loss. The adaptive consensus brings substantial improvement compared with DAdam. We believe further improvement is possible if we take the adaptive consensus into account when designing the optimizer (see the discussion in Appendix A.6).

## 5 CONCLUSION

This work challenges the long-standing perception that decentralized training inevitably sacrifices generalization for communication efficiency. Through DSGD-AC, we demonstrate that maintaining controlled consensus errors improves robustness and solution flatness, offering both practical scalability and superior model performance. The method introduces negligible computational overhead and integrates seamlessly with existing decentralized frameworks. Our experiments on CIFAR benchmarks and WMT14 confirm the broad applicability of this approach. These results suggest a paradigm shift: consensus errors should no longer be minimized at all costs but strategically managed as a form of implicit regularization. Beyond immediate applications to deep learning clusters, we envision that the principle of adaptive consensus could inform the design of future large-scale, resource-efficient, and generalizable learning systems. For a more extensive discussion about future potential improvements, please see Appendix A.6.

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

# A APPENDIX

## A.1 PROOF OF PROPOSITION 1

Recall that

$$
\begin{aligned}
P &= I - \tfrac{1}{n}\mathbf{1}\mathbf{1}^\top \\
L &= I - W = U_L \Lambda_L U_L^\top \\
\tilde{G}^{(t)} &= G^{(t)} P U_L \\
Z^{(t)} &= Z^{(t-1)}(I - \gamma^{(t)}\Lambda_L) - \alpha^{(t)}\tilde{G}^{(t)}
\end{aligned}
\tag{10}
$$

where $Z^{(t)}$ describes the consensus error projected onto the eigenbasis of the Laplacian.

Each column $z_k^{(t)}$ of $Z^{(t)}$ evolves as

$$
z_k^{(t)} = (1 - \gamma^{(t)}\lambda_k(L))z_k^{(t-1)} - \alpha^{(t)}\tilde{g}_k^{(t)}
\tag{11}
$$

where $\lambda_k(L)$ is the $k$-th eigenvalue of $L$.

To quantify the dynamics of $\|z_k^{(t)}\|_2^2$, consider a quasi-stationary regime where

$$
\mathbb{E}[\tilde{g}_i^{(t)}] = \mu_i, \quad \mathbb{E}[\|\tilde{g}_i^{(t)} - \mu_i\|_2^2] = \sigma_i^2
\tag{12}
$$

Then, by taking the expectation on Eq. (11) and letting $m_i = \mathbb{E}[z_i]$, we have

$$
m_i = (1 - \gamma^{(t)}\lambda_i(L))m_i - \alpha^{(t)}\mu_i
\tag{13}
$$

from which we find (for all modes $i \geq 2$)

$$
m_i = -\frac{1}{\lambda_i(L)}\frac{\alpha^{(t)}}{\gamma^{(t)}}\mu_i
\tag{14}
$$

Next, we define $\tilde{z}_i^{(t)} = z_i^{(t)} - m_i$ so that

$$
\tilde{z}_i^{(t)} = (1 - \gamma^{(t)}\lambda_i(L))\tilde{z}_i^{(t-1)} - \alpha^{(t)}(\tilde{g}_i^{(t-1)} - \mu_i)
$$

where we have subtracted $m_i$ from both sides and used the expression for $m_i$ just derived. Letting $V_i = \mathbb{E}[\|\tilde{z}_i^{(t)}\|_2^2]$ and assuming that the innovation $\eta^{(t-1)} = \tilde{g}_i^{(t-1)} - \mu_i$ is conditionally independent given all events up to iteration $t-1$, we obtain

$$
V_i = (1 - \gamma^{(t)}\lambda_i(L))^2 V_i + (\alpha^{(t)})^2\sigma_i^2.
$$

Solving for $V_i$ gives

$$
V_i = \frac{(\alpha^{(t)})^2}{1 - (1 - \gamma^{(t)}\lambda_i(L))^2}\sigma_i^2 = \frac{(\alpha^{(t)})^2}{2\lambda_i(L)\gamma^{(t)} - \lambda_i(L)^2(\gamma^{(t)})^2}\sigma_i^2.
$$

For $t$ large enough we have $\lambda_i(L)\gamma^{(t)} \leq 1$, so

$$
\lambda_i(L)\gamma^{(t)} \leq 2\lambda_i(L)\gamma^{(t)} - \lambda_i(L)^2(\gamma^{(t)})^2 \leq 2\lambda_i(L)\gamma^{(t)}.
$$

Consequently,

$$
\frac{(\alpha^{(t)})^2}{2\lambda_i(L)\gamma^{(t)}}\sigma_i^2 \leq V_i \leq \frac{(\alpha^{(t)})^2}{\lambda_i(L)\gamma^{(t)}}\sigma_i^2,
\tag{15}
$$

so in the quasi-stationary regime we have $V_i = \Theta\big((\alpha^{(t)})^2/\gamma^{(t)}\big)$. Combining (15) with (14) yields

$$
\mathbb{E}\left[\|z_i^{(t)}\|_2^2\right] = V_i + \|m_i\|_2^2 = V_i + \frac{(\alpha^{(t)})^2}{\lambda_i^2(\gamma^{(t)})^2}\|\mu_i\|_2^2.
$$

If $\gamma^{(t)} \geq \gamma_{\min} > 0$ and $\alpha^{(t)} \to 0$, then by Eq. (15) we have $V_i \leq (\alpha^{(t)})^2\sigma_i^2/(\lambda_i(L)\gamma_{\min}) \to 0$, and similarly $\|m_i\|_2^2 = O\big((\alpha^{(t)})^2\big) \to 0$. Hence $\mathbb{E}\left[\|z_i^{(t)}\|_2^2\right] \to 0$ for each $i$, so a constant consensus weight $\gamma^{(t)}$ cannot maintain a non-vanishing disagreement radius.

With the schedule $\gamma^{(t)} = g_0(\alpha^{(t)})^p$ and $g_0 > 0$, on the other hand, we obtain the lower bound

$$\mathbb{E}\left[\|z_i^{(t)}\|_2^2\right] \geq \frac{1}{\lambda_i(L)^2 g_0^2} (\alpha^{(t)})^{2-2p} \|\mu_i\|_2^2 + \frac{1}{2\lambda_i(L)g_0} (\alpha^{(t)})^{2-p} \sigma_i^2.$$

If $p \geq 2$, at least one of the exponents $2 - 2p$ or $2 - p$ is non-positive, so the right-hand side does not converge to zero as $\alpha^{(t)} \to 0$. Thus for $p \geq 2$ the per-mode energy $\mathbb{E}\left[\|z_i^{(t)}\|_2^2\right]$ is non-vanishing in the quasi-stationary regime. Finally, since $\|Z^{(t)}\|_F^2 = \|\Delta^{(t)}\|_F^2$ by orthogonality of $U$, a non-vanishing $\|Z^{(t)}\|_F^2$ implies that $r_t^2 = \mathbb{E}\|\Delta^{(t)}\|_F^2$ is non-vanishing as well.

## A.2 PROOF OF PROPOSITION 2

We work in a neighbourhood of a locally strongly convex minimizer $x^*$ of $F$, with Hessian $H = U_H \Lambda_H U_H^\top$ at $x^*$ and eigenpairs $\{(\lambda_k(H), u_k)\}_{k=1}^d$, $\lambda_k(H) > 0$. Let $N$ be the number of agents and collect the local iterates in

$$X^{(t)} := \left[x_1^{(t)}, \ldots, x_N^{(t)}\right] \in \mathbb{R}^{d \times N}, \qquad X^* := \left[x^*, \ldots, x^*\right] \in \mathbb{R}^{d \times N}.$$

We denote the communication matrix by $W$ (symmetric, doubly stochastic), and its Laplacian by $L = I - W$.

Let $G^{(t)} = [g_1^{(t)}, \ldots, g_N^{(t)}]$ denote the stochastic gradients used at time $t$, and define the gradient noise

$$\Xi^{(t)} := G^{(t)} - \nabla F(X^{(t-1)}), \qquad \nabla F(X^{(t-1)}) := \left[\nabla f_1(x_1^{(t-1)}), \ldots, \nabla f_N(x_N^{(t-1)})\right].$$

The DSGD-AC update can then be written as

$$X^{(t)} = X^{(t-1)}(I - \gamma^{(t)}L) - \alpha^{(t)}\left(\nabla F(X^{(t-1)}) + \Xi^{(t)}\right). \tag{16}$$

Let $P := I - \frac{1}{N}\mathbf{1}\mathbf{1}^\top$ be the projection onto the disagreement subspace, and define the consensus error matrix

$$\Delta^{(t)} := X^{(t)}P,$$

whose columns are precisely the disagreements $\delta_i^{(t)} = x_i^{(t)} - \bar{x}^{(t)}$. Using $LP = PL = L$ (since $L\mathbf{1} = 0$) and multiplying Eq. (16) on the right by $P$ yields

$$\Delta^{(t)} = \Delta^{(t-1)}(I - \gamma^{(t)}L) - \alpha^{(t)}\nabla F(X^{(t-1)})P - \alpha^{(t)}\Xi^{(t)}P. \tag{17}$$

By the i.i.d. local data assumption we have $f_i \equiv F$ for all $i$ and therefore $H_i(x^*) = H$ at the shared minimizer. A first-order Taylor expansion around $x^*$ gives, for each $i$,

$$\nabla f_i(x_i^{(t-1)}) = \nabla f_i(x^*) + H(x_i^{(t-1)} - x^*) + r_i^{(t-1)},$$

where the remainder $r_i^{(t-1)}$ is $O(\|x_i^{(t-1)} - x^*\|^2)$. At $x^*$ we have $\nabla f_i(x^*) = 0$, and in a sufficiently small neighbourhood of $x^*$ we may neglect the $r_i^{(t-1)}$ terms, which yields the local approximation

$$\nabla F(X^{(t-1)}) \approx H(X^{(t-1)} - X^*). \tag{18}$$

Since $X^*P = 0$, this implies

$$\nabla F(X^{(t-1)})P \approx H\Delta^{(t-1)}. \tag{19}$$

Substituting Eq. (19) into Eq. (17) gives the linearized consensus-error dynamics

$$\Delta^{(t)} \approx \Delta^{(t-1)}(I - \gamma^{(t)}L) - \alpha^{(t)}H\Delta^{(t-1)} - \alpha^{(t)}\Xi^{(t)}P. \tag{20}$$

We now project onto the eigenvectors of $H$. Let $U_H = [u_1, \ldots, u_d]$ collect the eigenvectors of $H$ and $\Lambda_H = \text{diag}(\lambda_1(H), \ldots, \lambda_d(H))$. For each $k$, define the projection of the consensus error onto $u_k$ by

$$\Delta_k^{(t)} := u_k^\top \Delta^{(t)} \in \mathbb{R}^{1 \times N},$$

and the projected noise

$$\xi_k^{(t)} := u_k^\top \Xi^{(t)} P \in \mathbb{R}^{1 \times N}.$$

Left-multiplying Eq. (20) by $u_k^\top$ and using $H u_k = \lambda_k(H) u_k$ yields

$$\Delta_k^{(t)} \approx \Delta_k^{(t-1)}(I - \gamma^{(t)} L) - \alpha^{(t)} \lambda_k(H)\, \Delta_k^{(t-1)} - \alpha^{(t)}\, \xi_k^{(t)}. \tag{21}$$

Thus, for each $k$, the projected consensus error $\Delta_k^{(t)}$ evolves as a linear system on $\mathbb{R}^N$ driven by noise $\xi_k^{(t)}$.

To study stability, we temporarily freeze the stepsizes on a short time window around a fixed time $t$, writing $\alpha = \alpha^{(t)}$ and $\gamma = \gamma^{(t)}$. Then Eq. (21) becomes

$$\Delta_k^{(t)} = \Delta_k^{(t-1)} A_k - \alpha\, \xi_k^{(t)}, \qquad A_k := I - \gamma L - \alpha \lambda_k(H) I.$$

Using $L = I - W$, we can rewrite $A_k$ as

$$A_k = I - \gamma(I - W) - \alpha \lambda_k(H) I = \big(1 - \gamma - \alpha \lambda_k(H)\big) I + \gamma W. \tag{22}$$

Since $W$ is symmetric and doubly stochastic, its eigenvalues $\{\lambda_j(W)\}_{j=1}^N$ lie in $(-1, 1]$, with $\lambda_1(W) = 1$ because the graph is connected. The eigenvalues of $A_k$ are therefore

$$\mu_{k,j} = 1 - \gamma - \alpha \lambda_k(H) + \gamma \lambda_j(W), \qquad j = 1, \dots, N.$$

As $A_k$ is symmetric, mean-square stability of the homogeneous system $\Delta_k^{(t)} = \Delta_k^{(t-1)} A_k$ is equivalent to $|\mu_{k,j}| < 1$ for all $j$. Thus we require

$$-1 < 1 - \gamma - \alpha \lambda_k(H) + \gamma \lambda_j(W) < 1 \quad \forall j.$$

The right inequality is automatically satisfied for $\alpha > 0$ because $\lambda_1(W) = 1$ and $\lambda_j(W) \le 1$ imply

$$1 - \gamma - \alpha \lambda_k(H) + \gamma \lambda_j(W) \le 1 - \alpha \lambda_k(H) < 1.$$

The left inequality is most restrictive for the smallest eigenvalue $\lambda_{\min}^W$ of $W$, giving

$$1 - \gamma - \alpha \lambda_k(H) + \gamma \lambda_{\min}(W) > -1 \quad \Longleftrightarrow \quad \alpha < \frac{2 + (\lambda_{\min}(W) - 1)\gamma}{\lambda_k(H)}.$$

This is exactly the condition Eq. (9).

Since the right-hand side of Eq. (9) is strictly decreasing in $\lambda_k(H)$, non-increasing stepsizes $\alpha^{(t)}$ will enter the stable regime for small-curvature modes $k$ earlier along the training trajectory, while high-curvature modes remain closer to instability for longer. In the presence of comparable injected noise, these high-curvature modes therefore sustain larger steady-state variance, which completes the proof.

### A.3 LOSS ENVELOPE AND CURVATURE TILT

In this section we relate the consensus errors maintained by DSGD-AC to the local geometry of the global objective. Our goal is to understand which perturbations of the deployed model $\bar{x}(t)$ are implicitly favored or suppressed by the algorithm.

Lemma 1 shows that, in a neighbourhood of a locally strongly convex minimizer $x^*$, the average local loss decomposes into the loss at the deployed model plus a quadratic envelope term depending on the disagreement covariance $\Sigma_t$, up to higher-order corrections. Thus, in this regime, DSGD-AC can be viewed as minimizing $F(\bar{x}(t))$ together with a Hessian-weighted disagreement penalty.

**Lemma 1 (Local loss envelope)** *Let $x^*$ be a locally strongly convex minimizer of $F$ with Hessian $H$ at $x^*$. For a fixed time $t$, let $\bar{x}(t) := \frac{1}{n} \sum_{i=1}^n x_i^{(t)}$ be the deployed model and define the disagreements $e_i^{(t)} := x_i^{(t)} - \bar{x}(t)$. Let*

$$\Sigma_t := \frac{1}{n} \sum_{i=1}^n e_i^{(t)} e_i^{(t)^\top}$$

*denote their empirical covariance. Assume that $\|x_i^{(t)} - x^*\|$ is small for all $i$. Then*

$$\frac{1}{n} \sum_{i=1}^n f_i\big(x_i^{(t)}\big) = F(\bar{x}(t)) + \tfrac{1}{2} \operatorname{tr}(H \Sigma_t) + O\big((\operatorname{tr} \Sigma_t)^{3/2}\big). \tag{23}$$

*Proof.* Fix $t$ and abbreviate $\bar{x} = \bar{x}^{(t)}$, $e_i = e_i^{(t)}$. A Taylor expansion of $f_i$ around $\bar{x}$ yields

$$f_i(\bar{x} + e_i) = f_i(\bar{x}) + \nabla f_i(\bar{x})^\top e_i + \tfrac{1}{2} e_i^\top H e_i + R_i,$$

where $R_i = O(\|e_i\|^3)$ and we used that all $f_i$ have Hessian $H$ at $x^*$. Averaging over $i$ gives

$$\frac{1}{n} \sum_{i=1}^n f_i(x_i^{(t)}) = F(\bar{x}) + \frac{1}{n} \sum_{i=1}^n \nabla f_i(\bar{x})^\top e_i + \frac{1}{2n} \sum_{i=1}^n e_i^\top H e_i + \frac{1}{n} \sum_{i=1}^n R_i.$$

By definition of $\bar{x}$ we have $\sum_{i=1}^n \delta_i = 0$, hence

$$\frac{1}{n} \sum_{i=1}^n \nabla f_i(\bar{x})^\top e_i = \nabla F(\bar{x})^\top \Big( \frac{1}{n} \sum_{i=1}^n e_i \Big) = 0.$$

Moreover,

$$\frac{1}{2n} \sum_{i=1}^n e_i^\top H e_i = \frac{1}{2} \operatorname{tr}\bigg( H \frac{1}{n} \sum_{i=1}^n e_i e_i^\top \bigg) = \tfrac{1}{2} \operatorname{tr}(H \Sigma_t).$$

For the remainder, there exists a constant $C > 0$ such that $|R_i| \le C \|\delta_i\|^3$ in the local region. Thus

$$\Big| \frac{1}{n} \sum_{i=1}^n R_i \Big| \le \frac{C}{n} \sum_{i=1}^n \|e_i\|^3 \le C \Big( \frac{1}{n} \sum_{i=1}^n \|e_i\|^2 \Big)^{3/2} = C \big( \operatorname{tr} \Sigma_t \big)^{3/2},$$

where the second step follows from Hölder's inequality. This gives the claimed $O\big((\operatorname{tr} \Sigma_t)^{3/2}\big)$ bound and for the remainder and concludes the proof. $\qquad\square$

To understand how this disagreement penalty depends on curvature and on the communication graph, we diagonalize the local dynamics in the joint eigenbasis of the Hessian and the Laplacian. This leads to the following spectral representation.

**Proposition 3 (Curvature tilt)** *Under the assumptions and notation of Proposition 2 and Lemma 1, fix a time $t$ in a local quasi-stationary regime and freeze $\alpha = \alpha^{(t)}$ and $\gamma = \gamma^{(t)}$. Let $L = I - W$ be the graph Laplacian and denote its eigenvalues by $0 = \lambda_1(L) < \lambda_2(L) \le \cdots \le \lambda_N(L)$, and let $\lambda_1(H) \le \cdots \le \lambda_d(H)$ be the eigenvalues of $H$. Let $\Sigma_t$ be the disagreement covariance at time $t$. Then the leading-order Hessian-weighted disagreement envelope can be written as*

$$\frac{1}{2} \mathbb{E}\big[\operatorname{tr}(H \Sigma_t)\big] \approx \frac{(\alpha^{(t)})^2}{4N} \sum_{j=2}^N \sum_{k=1}^d w_j\big(\lambda_k(H)\big) q_{k,j}, \tag{25}$$

*where $q_{k,j} \ge 0$ is the innovation variance of the Laplacian–Hessian mode $(j,k)$ and, for $\lambda \ge 0$,*

$$w_j(\lambda) := \frac{\lambda}{\gamma^{(t)} \lambda_j(L) + \alpha^{(t)} \lambda}. \tag{26}$$

*For each fixed graph mode $j \ge 2$, the weight $w_j(\lambda)$ is strictly increasing in $\lambda$.*

*Proof.* We work in the local quadratic regime around $x^*$ and on a short time window around $t$ where we freeze $\alpha = \alpha^{(t)}$ and $\gamma = \gamma^{(t)}$. From the linearization in Appendix A.2 (cf. the proof of Proposition 2) we have

$$\Delta^{(s+1)} \approx \Delta^{(s)}(I - \gamma L) - \alpha H \Delta^{(s)} - \alpha \, \Xi^{(s+1)} P, \tag{24}$$

for $s$ in a short window around $t$, where $\Xi^{(s+1)}$ collects the gradient noise and $P$ is the projection onto the disagreement subspace.

Let $L = U_L \Lambda_L U_L^\top$ and $H = U_H \Lambda_H U_H^\top$ be the eigendecompositions of the Laplacian and Hessian, with eigenvalues $0 = \lambda_1(L) < \lambda_2(L) \le \cdots \le \lambda_n(L)$ and $0 < \lambda_1(H) \le \cdots \le \lambda_d(H)$. We write the consensus error in the joint eigenbasis as

$$\Delta^{(s)} = U_H Y^{(s)} U_L^\top,$$

for some coefficient matrices $Y^{(s)} \in \mathbb{R}^{d \times N}$, and define the corresponding noise coefficients

$$Z^{(s+1)} := U_H^\top \Xi^{(s+1)} P U_L.$$

Substituting these into Eq. (24) and using $I - \gamma L = U_L(I - \gamma \Lambda_L)U_L^\top$ and $H = U_H \Lambda_H U_H^\top$ gives

$$Y^{(s+1)} = Y^{(s)}(I - \gamma \Lambda_L) - \alpha \Lambda_H Y^{(s)} - \alpha Z^{(s+1)}.$$

Taking the $(k, j)$ entry yields, for $k = 1, \dots, d$ and $j = 1, \dots, N$,

$$Y_{k,j}^{(s+1)} = a_{k,j} Y_{k,j}^{(s)} - \alpha \zeta_{k,j}^{(s+1)}, \qquad a_{k,j} := 1 - \gamma \lambda_j(L) - \alpha \lambda_k(H), \tag{25}$$

where $\zeta_{k,j}^{(s+1)} := Z_{k,j}^{(s+1)}$. Since $\Delta^{(s)}$ lies in the disagreement subspace, the consensus graph mode $j = 1$ does not contribute and we may restrict to $j \geq 2$.

On the short time window around $t$, we approximate Eq. (25) as a stationary AR(1) recursion driven by zero-mean innovations with variance

$$q_{k,j} := \mathrm{Var}(\zeta_{k,j}^{(s)}).$$

Assuming $|a_{k,j}| < 1$ (the stability condition of Proposition 2) and that the innovations are uncorrelated across the short time window, the stationary variance $S_{k,j} := \mathrm{Var}(Y_{k,j})$ satisfies the scalar Lyapunov equation

$$S_{k,j} = a_{k,j}^2 S_{k,j} + \alpha^2 q_{k,j},$$

hence

$$S_{k,j} = \frac{\alpha^2}{1 - a_{k,j}^2} q_{k,j}. \tag{26}$$

Using $a_{k,j} = 1 - \gamma \lambda_j(L) - \alpha \lambda_k(H)$, we compute

$$1 - a_{k,j}^2 = 1 - \big(1 - \gamma \lambda_j(L) - \alpha \lambda_k(H)\big)^2 = 2(\gamma \lambda_j(L) + \alpha \lambda_k(H)) - (\gamma \lambda_j(L) + \alpha \lambda_k(H))^2.$$

In the small-stepsize regime where $\gamma \lambda_j(L) + \alpha \lambda_k(H)$ is small, the quadratic term can be neglected and we obtain the approximation

$$S_{k,j} \approx \frac{\alpha^2}{2(\gamma \lambda_j(L) + \alpha \lambda_k(H))} q_{k,j}. \tag{27}$$

Next, recall that

$$\Sigma_t = \frac{1}{n} \mathbb{E}\big[\Delta^{(t)} \Delta^{(t)\top}\big].$$

Using $\Delta^{(t)} = U_H Y^{(t)} U_L^\top$ and orthogonality of $U_H$ and $U_L$, we obtain

$$\mathbb{E}\big[\mathrm{tr}(H \Sigma_t)\big] = \frac{1}{n} \mathbb{E}\big[\mathrm{tr}(H \Delta^{(t)} \Delta^{(t)\top})\big] = \frac{1}{n} \mathbb{E}\big[\mathrm{tr}(\Lambda_H Y^{(t)} Y^{(t)\top})\big].$$

The last trace equals $\sum_{k=1}^d \lambda_k(H) \sum_{j=1}^n \mathbb{E}[(Y_{k,j}^{(t)})^2]$. Approximating $\mathbb{E}[(Y_{k,j}^{(t)})^2]$ by the stationary variance $S_{k,j}$ in Eq. (27) for $j \geq 2$ and noting again that the $j = 1$ consensus mode does not contribute, we obtain

$$\mathbb{E}\big[\mathrm{tr}(H \Sigma_t)\big] \approx \frac{\alpha^2}{2n} \sum_{j=2}^n \sum_{k=1}^d \frac{\lambda_k(H)}{\gamma \lambda_j(L) + \alpha \lambda_k(H)} q_{k,j}.$$

Multiplying by $\frac{1}{2}$ yields

$$\frac{1}{2} \mathbb{E}\big[\mathrm{tr}(H \Sigma_t)\big] \approx \frac{\alpha^2}{4n} \sum_{j=2}^n \sum_{k=1}^d w_j(\lambda_k(H)) q_{k,j},$$

with

$$w_j(\lambda) := \frac{\lambda}{\gamma \lambda_j(L) + \alpha \lambda},$$

which is exactly Eq. (25)–Eq. (26).

Finally, for each fixed $j \geq 2$ we have $\lambda_j(L) > 0$ and $\alpha, \gamma > 0$, so for $\lambda \geq 0$,

$$w_j'(\lambda) = \frac{\gamma \lambda_j(L)}{\left(\gamma \lambda_j(L) + \alpha \lambda\right)^2} > 0.$$

Thus $w_j(\lambda)$ is strictly increasing in $\lambda$ for every $j \geq 2$, which completes the proof. $\square$

The spectral form in Proposition 3 separates the envelope into curvature-dependent weights $w_j(\lambda_k(H))$ and mode-wise innovation variances $q_{k,j}$. To go further, we specialize to the case where these innovations arise from mini-batch SGD noise. A growing body of empirical and theoretical work has shown that, near a local minimum, the covariance of mini-batch SGD gradients is approximately Hessian-aligned and scales with both the loss value and curvature, $\mathrm{Cov}(g(x) - \nabla F(x)) \approx c_t L(x) H(x)$, in linear models and deep networks (e.g., Ziyin et al. (2022); Wu et al. (2022); Mori et al. (2022)). Under this structure the $q_{k,j}$ inherit the same dependence on the Hessian eigenvalues, which yields a curvature-dependent spectral penalty of the form in Eq. (29).

**Corollary 1 (Hessian-aligned mini-batch noise)** *Under the assumptions and notation of Proposition 3, assume in addition that the gradient noise driving DSGD-AC is inherited from a mini-batch SGD oracle whose covariance is approximately Hessian-aligned,*

$$\mathrm{Cov}\big(g_i(x) - \nabla f_i(x)\big) \approx c_t\, L(x)\, H(x), \tag{28}$$

*for some scalar factor $c_t > 0$ depending on the batch size and possibly on time $t$. Then, in the local quadratic regime around $x^*$, the leading-order Hessian-weighted disagreement envelope can be written as*

$$\frac{1}{2}\, \mathbb{E}\big[\mathrm{tr}(H\Sigma_t)\big] \approx L(\bar{x}^{(t)}) \sum_{k=1}^{d} \omega_t\big(\lambda_k(H)\big), \tag{29}$$

*where $\omega_t : [0,\infty) \to [0,\infty)$ is strictly increasing and satisfies that $\lambda \mapsto \omega_t(\lambda)/\lambda$ is also strictly increasing on $(0,\infty)$. In particular, larger Hessian eigenvalues receive a disproportionately larger penalty relative to their magnitude than smaller ones.*

*Proof.* By Proposition 3, the leading-order envelope can be written as

$$\frac{1}{2}\, \mathbb{E}\big[\mathrm{tr}(H\Sigma_t)\big] \approx \frac{(\alpha^{(t)})^2}{4n} \sum_{j=2}^{n} \sum_{k=1}^{d} w_j\big(\lambda_k(H)\big)\, q_{k,j},$$

with

$$w_j(\lambda) = \frac{\lambda}{\gamma^{(t)} \lambda_j(L) + \alpha^{(t)} \lambda},$$

and $q_{k,j}$ the innovation variances of the joint Laplacian–Hessian modes. Under the Hessian-aligned covariance structure Eq. (28), the per-step gradient noise covariance in the Hessian eigenbasis is approximately diagonal with entries proportional to $L(\bar{x}^{(t)}) \lambda_k(H)$. Projecting into the joint basis, the $q_{k,j}$ inherit this alignment and, up to graph-dependent constants, satisfy

$$q_{k,j} \approx c_t\, L(\bar{x}^{(t)})\, \lambda_k(H).$$

Substituting this scaling gives

$$\frac{1}{2}\, \mathbb{E}\big[\mathrm{tr}(H\Sigma_t)\big] \approx L(\bar{x}^{(t)})\, \frac{(\alpha^{(t)})^2 c_t}{4N} \sum_{j=2}^{N} \sum_{k=1}^{d} \frac{\lambda_k(H)^2}{\gamma^{(t)} \lambda_j(L) + \alpha^{(t)} \lambda_k(H)}.$$

and we recover (29). with

$$\omega_t(\lambda) := \frac{(\alpha^{(t)})^2 c_t}{4N} \sum_{j=2}^{N} \frac{\lambda^2}{\gamma^{(t)} \lambda_j(L) + \alpha^{(t)} \lambda}.$$

It remains to verify the monotonicity properties of $\omega_t$. For each fixed $j \geq 2$, define

$$h_{t,j}(\lambda) := \frac{\lambda^2}{\gamma^{(t)} \lambda_j(L) + \alpha^{(t)} \lambda} = \lambda\, w_j(\lambda),$$

where $w_j(\lambda)$ is the weight from Proposition 3. We have already shown that, for $\lambda \geq 0$, $w_j(\lambda) \geq 0$ and $w_j(\lambda)$ is strictly increasing. Therefore, for $\lambda > 0$,

$$h'_{t,j}(\lambda) = w_j(\lambda) + \lambda w'_j(\lambda) > 0,$$

so each $h_{t,j}$ is strictly increasing on $(0, \infty)$ (and nondecreasing on $[0, \infty)$). Since $\omega_t$ is a positive linear combination of the $h_{t,j}$, $\omega_t(\lambda)$ is strictly increasing on $(0, \infty)$. Moreover,

$$\frac{\phi_t(\lambda)}{\lambda} = \frac{(\alpha^{(t)})^2 c_t}{4n} \sum_{j=2}^{n} \frac{\lambda}{\gamma^{(t)} \lambda_j(L) + \alpha^{(t)} \lambda} = \frac{(\alpha^{(t)})^2 c_t}{4n} \sum_{j=2}^{n} w_j(\lambda).$$

Each $w_j(\lambda)$ is strictly increasing in $\lambda$ by Proposition 3, so their positive linear combination $\omega_t(\lambda)/\lambda$ is also strictly increasing on $(0, \infty)$. This proves the monotonicity stated in the corollary. □

**Remark** The spectral form in Eq. (29) shows that, under Hessian-aligned mini-batch noise, the leading-order loss inflation induced by DSGD-AC behaves like an implicit spectral penalty of the form $L(\bar{x}(t)) \sum_k \omega_t(\lambda_k(H))$, where both $\omega_t(\lambda)$ and $\omega_t(\lambda)/\lambda$ are strictly increasing. In particular, larger eigenvalues of $H$ are penalized disproportionately more per unit curvature than smaller ones. This contrasts with classical criteria that depend only on $\mathrm{tr}(H)$ or $\log \det H$, and implies that the top eigenvalues of $H$ are implicitly regularized by the combination of consensus noise and mini-batch SGD. Conceptually, this connects DSGD-AC to explicit eigenvalue regularization schemes that aim to control large curvature directions in sharpness-aware methods such as Eigen-SAM (Luo et al., 2024) or Hessian-based noise-stability regularization (Zhang et al., 2023), but here the regularization arises automatically from decentralized averaging and stochastic gradients rather than from additional optimization steps.

### A.4 EXPERIMENT DETAILS

#### A.4.1 IMAGE CLASSIFICATION EXPERIMENTS ON CIFAR10

The selection of hyperparameters follows the original paper (Zagoruyko & Komodakis, 2016), and our baseline implementation perfectly matches its performance.

| Category | Setting |
|---|---|
| *General* | |
| Number of epochs | 200 |
| Global batch size | 128 |
| Learning rate scheduler | Linear warm-up to $0.1$ in the first 10 epochs, followed by cosine annealing until the end |
| Base optimizer | SGD with momentum ($0.9$), weight decay $= 5 \times 10^{-4}$ |
| Data shuffle | Randomly shuffled and split into $N$ local datasets each epoch |
| *Decentralized training* | |
| Number of workers | 8 |
| Communication topology | One-peer ring (alternating between neighbors $i - 1$ and $i + 1$ across iterations) |
| DSGD-AC parameters | Exponent $p = 3$ (tuned based on experiments); $\gamma = 1$ during warm-up |
| BatchNorm calibration | Similar to the case in (Defazio et al., 2024), a calibration on the BatchNorm statistics is needed because there is a mismatch between the local models and the global average. To calibrate mismatched statistics, a full pass over the training set is conducted before validation. Only one calibration should be done if intermediate checkpoints are not evaluated. |

#### A.4.2 TRANSFORMER ON WMT14

The selection of hyperparameters follows the original paper (Vaswani et al., 2017), and our baseline implementation perfectly matches its performance.

| Category | Setting |
|---|---|
| ***General*** | |
| Number of epochs | 20 |
| Global batch size | $\sim$25k tokens |
| Learning rate scheduler | Linear warm-up to $5 \times 10^{-4}$ over the first 4000 iterations, then decay as $\alpha_0 \cdot (4000/t)^{0.5}$ ($t$ is the iteration index). $\alpha_0 = 0.0005$ for centralized Adam, and $\alpha_0 = 0.0013$ for decentralized methods. |
| Base optimizer | Adam ($\beta_1 = 0.9, \beta_2 = 0.98$) for centralized Adam, and ($\beta_1 = 0.974$, $\beta_2 = 0.999$) for decentralized methods. |
| Data shuffle | Randomly shuffled and split into $N$ local datasets each epoch |
| ***Decentralized training*** | |
| Number of workers | 8 |
| Communication topology | One-peer ring (alternating between neighbors $i - 1$ and $i + 1$ across iterations) |
| DSGD-AC parameters | Exponent $p = 2$ (tuned based on experiments); $\gamma = 1$ during warm-up |
| Normalization | Since only layer normalization is used, no calibration is needed. |

### A.5 ADDITIONAL EXPERIMENT RESULTS

#### A.5.1 IMAGE CLASSIFICATION WITH WIDE RESNET

The complete statistics of the image classification task are deferred to this section due to the space limit. Even though comparing DSGD-AC with SAM-like methods is not our emphasis, we implement SAM (Foret et al., 2020) and the average-direction SAM (Bisla et al., 2022) and report their results for reference. We follow Foret et al. (2020) to use $\rho = 0.05$ in all the experiments, and use the same schedule of the variance of the random perturbations as described in the official GitHub repository[1] (Bisla et al., 2022).

Figures 5, 6, 7, and 8 and Table 3 present all the results on the image classification task. We summary the results as

- SAM always outperforms other methods at the cost of $2\times$ computation.

- DSGD-AC always achieves the best test loss among the methods with $1\times$ computation.

- AD-SAM outperforms DSGD-AC in the solution flatness only on experiments with WRN16-8, which is relatively smaller than WRN28-10.

Note that (1) for training loss, it is evaluated on the workers for decentralized training, and evaluated on perturbed points for SAM, (2) for test loss, it's evaluated on the global average model for decentralized training, (3) each curve for each algorithm is based on 3 runs with the same set of random seeds, and (4) the shaded parts correspond to the 95% confidence interval.

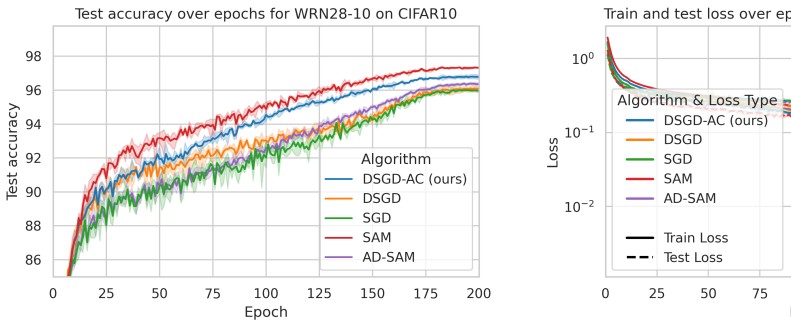

Figure 5: WRN28-10 on CIFAR-10. **Left**: Test accuracy on test set. For decentralized training, the accuracy is evaluated on the global average model. **Right**: Training and test losses.

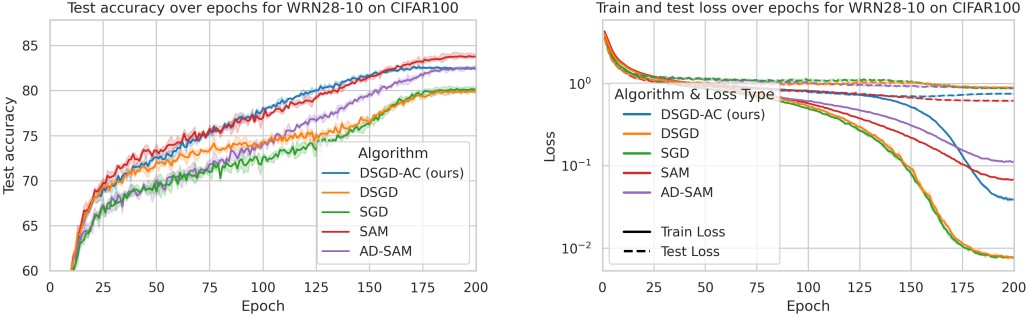

Figure 6: WRN28-10 on CIFAR-100. **Left**: Test accuracy on test set. For decentralized training, the accuracy is evaluated on the global average model. **Right**: Training and test losses.

---

[1] https://github.com/devansh20la/LPF-SGD/blob/master/codes/README.md

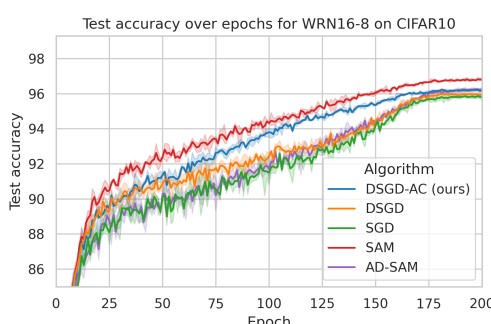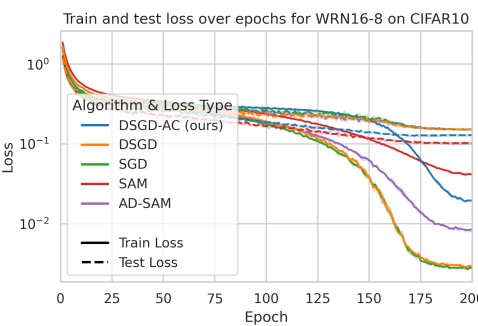

Figure 7: WRN16-8 on CIFAR-10. **Left**: Test accuracy on test set. For decentralized training, the accuracy is evaluated on the global average model. **Right**: Training and test losses.

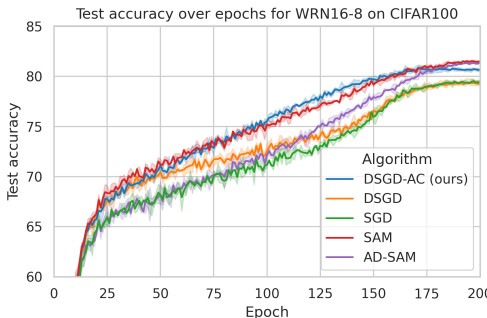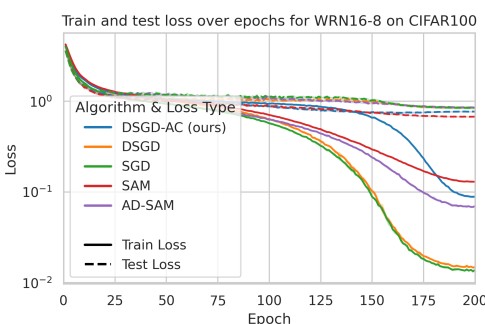

Figure 8: WRN16-8 on CIFAR-100. **Left**: Test accuracy on test set. For decentralized training, the accuracy is evaluated on the global average model. **Right**: Training and test losses.

| Model | Dataset | Algorithm | Test Acc. (%) ↑ | Test Loss ↓ | Mean Top-1 Eigenvalue ↓ | Computation ↓ |
|-------|---------|-----------|-----------------|-------------|--------------------------|---------------|
| WRN28-10 | CIFAR-10 | DSGD | 96.07 ± 0.13 | 0.176 ± 0.005 | 22.4360 ± 3.9916 | 1x |
| | | SGD | 95.96 ± 0.14 | 0.182 ± 0.004 | 16.8485 ± 0.3251 | 1x |
| | | DSGD-AC | 96.77 ± 0.11 | 0.128 ± 0.003 | 8.9693 ± 0.3514 | 1x |
| | | AD-SAM | 96.37 ± 0.11 | 0.168 ± 0.002 | 24.9059 ± 1.6212 | 1x |
| | | SAM | **97.33 ± 0.04** | **0.100 ± 0.002** | **0.3523 ± 0.0312** | 2x |
| | CIFAR-100 | DSGD | 79.86 ± 0.22 | 0.899 ± 0.008 | 49.5719 ± 4.8022 | 1x |
| | | SGD | 80.15 ± 0.42 | 0.878 ± 0.020 | 37.3799 ± 2.8886 | 1x |
| | | DSGD-AC | 82.38 ± 0.09 | 0.755 ± 0.008 | 19.8061 ± 0.6653 | 1x |
| | | AD-SAM | 82.57 ± 0.31 | 0.891 ± 0.007 | 32.6371 ± 2.3362 | 1x |
| | | SAM | **83.79 ± 0.25** | **0.618 ± 0.003** | **1.7295 ± 0.0385** | 2x |
| WRN16-8 | CIFAR-10 | DSGD | 95.94 ± 0.11 | 0.152 ± 0.001 | 18.1998 ± 0.6427 | 1x |
| | | SGD | 95.81 ± 0.13 | 0.153 ± 0.003 | 17.4934 ± 1.6191 | 1x |
| | | DSGD-AC | 96.17 ± 0.04 | 0.129 ± 0.003 | 11.8250 ± 0.4883 | 1x |
| | | AD-SAM | 96.25 ± 0.12 | 0.152 ± 0.002 | 8.5178 ± 0.5453 | 1x |
| | | SAM | **96.81 ± 0.08** | **0.102 ± 0.003** | **1.3928 ± 0.0586** | 2x |
| | CIFAR-100 | DSGD | 79.25 ± 0.26 | 0.854 ± 0.016 | 36.1998 ± 3.8028 | 1x |
| | | SGD | 79.42 ± 0.18 | 0.849 ± 0.015 | 33.7733 ± 0.7897 | 1x |
| | | DSGD-AC | 80.67 ± 0.11 | 0.771 ± 0.005 | 19.8032 ± 0.1652 | 1x |
| | | AD-SAM | 81.36 ± 0.06 | 0.858 ± 0.004 | 17.5450 ± 1.2583 | 1x |
| | | SAM | **81.51 ± 0.08** | **0.677 ± 0.003** | **4.7932 ± 0.1957** | 2x |

Table 3: Algorithm comparison on image classification including SAM (Foret et al., 2020) and average-direction SAM (Bisla et al., 2022).

### A.5.2 VARYING NUMBER OF WORKERS AND TOPOLOGY

In this section, we evalute the performance of DSGD-AC with various number of workers (8, 16, 32) and with various communication topologies (one-peer ring, exponential graph (Ying et al., 2021), complete graph). The results where DSGD-AC outperforms centralized SGD are marked in green

and those where it under-performs are marked in red . As in the main text, the statistics of each result are based on 3 random runs. For the 32-worker case, we used 256 as the global batch size for an appropriate scaling efficiency, and the learning rate is linearly scaled up by $2\times$ accordingly.

**Interpretation of the results**

- The results in Tables 4, 5 and 6 demonstrate a superior performance of DSGD-AC over DSGD and SGD with 8 and 16 workers on almost all topologies (except WRN16-8 on CIFAR10 with 16 workers and one-peer ring topology).

- For the experiments with 32 workers, DSGD-AC does not bring further improvement compared with the DSGD baseline. This is expected as the variance in the updates is too large and, in this case, the adaptive consensus mechanism may hurt the convergence.

- To further improve the performance, we vary the epoch at which we activate the adaptive consensus in Section A.5.3. The results of DSGD-AC after tuning the start epoch, shown in Tables 7 and 8, demonstrate that this technique largely alleviates the observed problems and demonstrates the practicality of AC mechanism.

| n | topology | **Algorithms** (Test Acc. ↑ / Test Loss ↓) | |
| --- | --- | --- | --- |
| | | DSGD | DSGD-AC |
| 8 | complete | 79.66 ± 0.86 / 1.017 ± 0.027 | 81.83 ± 0.21 / 0.904 ± 0.028 |
| | exp | 80.03 ± 0.98 / 0.941 ± 0.023 | 81.99 ± 0.14 / 0.882 ± 0.027 |
| | ring | 79.86 ± 0.22 / 0.899 ± 0.008 | 82.38 ± 0.09 / 0.755 ± 0.008 |
| 16 | complete | 79.77 ± 0.35 / 1.080 ± 0.029 | 82.27 ± 0.44 / 0.768 ± 0.012 |
| | exp | 79.87 ± 0.36 / 0.983 ± 0.078 | 82.37 ± 0.36 / 0.753 ± 0.022 |
| | ring | 79.84 ± 1.17 / 0.964 ± 0.047 | 82.29 ± 0.03 / 0.748 ± 0.007 |
| 32 | complete | 79.90 ± 0.51 / 1.046 ± 0.045 | 81.86 ± 0.40 / 0.707 ± 0.005 |
| | exp | 80.38 ± 0.28 / 0.980 ± 0.056 | 80.50 ± 0.24 / 0.766 ± 0.017 |
| | ring | 80.52 ± 0.45 / 0.948 ± 0.042 | 78.56 ± 0.30 / 0.831 ± 0.010 |

Table 4: WRN28-10 on CIFAR100. Centralized SGD baseline: 80.15 ± 0.42 / 0.878 ± 0.020.

| n | topology | Algorithms (Test Acc. ↑ / Test Loss ↓) | |
|---|---|---|---|
| | | DSGD | DSGD-AC |
| 8 | complete | 95.90 ± 0.27 / 0.189 ± 0.003 | 96.48 ± 0.24 / 0.129 ± 0.006 |
| | exp | 96.08 ± 0.26 / 0.186 ± 0.016 | 96.61 ± 0.08 / 0.126 ± 0.003 |
| | ring | 96.07 ± 0.13 / 0.176 ± 0.005 | 96.77 ± 0.11 / 0.128 ± 0.003 |
| 16 | complete | 96.00 ± 0.53 / 0.179 ± 0.018 | 96.56 ± 0.26 / 0.115 ± 0.005 |
| | exp | 95.98 ± 0.18 / 0.194 ± 0.001 | 96.39 ± 0.14 / 0.116 ± 0.001 |
| | ring | 95.89 ± 0.42 / 0.196 ± 0.013 | 96.24 ± 0.18 / 0.117 ± 0.003 |
| 32 | complete | 95.87 ± 0.31 / 0.200 ± 0.013 | 96.01 ± 0.10 / 0.122 ± 0.002 |
| | exp | 95.88 ± 0.09 / 0.194 ± 0.008 | 95.27 ± 0.33 / 0.141 ± 0.005 |
| | ring | 96.16 ± 0.10 / 0.180 ± 0.002 | 94.44 ± 0.21 / 0.170 ± 0.007 |

Table 5: WRN28-10 on CIFAR10. Centralized SGD baseline: 95.96 ± 0.14 / 0.182 ± 0.004.

| n | topology | Algorithms (Test Acc. ↑ / Test Loss ↓) | |
|---|---|---|---|
| | | DSGD | DSGD-AC |
| 8 | complete | 95.82 ± 0.17 / 0.166 ± 0.008 | 96.22 ± 0.15 / 0.127 ± 0.003 |
| | exp | 95.68 ± 0.19 / 0.165 ± 0.008 | 96.19 ± 0.34 / 0.125 ± 0.010 |
| | ring | 95.94 ± 0.11 / 0.152 ± 0.001 | 96.17 ± 0.04 / 0.129 ± 0.003 |
| 16 | complete | 95.81 ± 0.25 / 0.157 ± 0.007 | 96.21 ± 0.16 / 0.115 ± 0.003 |
| | exp | 95.67 ± 0.11 / 0.162 ± 0.008 | 95.93 ± 0.11 / 0.122 ± 0.004 |
| | ring | 95.86 ± 0.31 / 0.161 ± 0.003 | 95.77 ± 0.21 / 0.125 ± 0.002 |
| 32 | complete | 95.77 ± 0.16 / 0.172 ± 0.002 | 95.56 ± 0.34 / 0.134 ± 0.002 |
| | exp | 95.76 ± 0.09 / 0.159 ± 0.011 | 95.03 ± 0.26 / 0.147 ± 0.003 |
| | ring | 95.65 ± 0.30 / 0.163 ± 0.012 | 94.24 ± 0.08 / 0.172 ± 0.006 |

Table 6: WRN16-8 on CIFAR10. Centralized SGD baseline: 95.81 ± 0.13 / 0.153 ± 0.003.

### A.5.3 VARYING EPOCH TO ENABLE ADAPTIVE CONSENSUS

In this section, we vary the epoch in which we enable the adaptive consensus (AC). In all experiment so far, we enabled AC directly after the warmup phase (10th epoch). We now test the performance of DSGD-AC with starting epochs {10 (*default*), 50, 100, 170, 200 (*equivalent to DSGD*)}, for different worker counts {16, 32} and topologies {complete, exponential graph, one-peer ring}.

In the experiments, we fix $p = 3$, and $\alpha_{\max}$ is taken as the learning rate at the start of the epoch when AC is enabled ($\alpha$ is monotonically decreasing after the warmup, so $\gamma$ is always kept in $(0, 1]$).

**Interpretation of the results**

- For the case with larger number of workers (32), delaying the activation of AC can bring better performance compared to the default setup.
- As shown in Tables 7 and 8, DSGD-AC can achieve both better test accuracy and better test loss than DSGD and (centralized) SGD on at least one starting epoch (that is not 200) on all setups. This implies that the AC mechanism can improve the generalization.

| Model / Dataset | Topology | Algorithm | |
|---|---|---|---|
| | | DSGD | DSGD-AC |
| WRN28-10 / CIFAR-100 | complete | 79.77 ± 0.35 / 1.080 ± 0.029 | **82.30 ± 0.17 / 0.791 ± 0.012** |
| | exp | 79.87 ± 0.36 / 0.983 ± 0.078 | **82.42 ± 0.20 / 0.760 ± 0.011** |
| | ring | 79.84 ± 1.17 / 0.964 ± 0.047 | **82.50 ± 0.33 / 0.756 ± 0.022** |
| WRN28-10 / CIFAR-10 | complete | 96.00 ± 0.53 / 0.179 ± 0.018 | **96.56 ± 0.21 / 0.115 ± 0.003** |
| | exp | 95.98 ± 0.18 / 0.194 ± 0.001 | **96.65 ± 0.16 / 0.122 ± 0.003** |
| | ring | 95.89 ± 0.42 / 0.196 ± 0.013 | **96.58 ± 0.18 / 0.121 ± 0.002** |
| WRN16-8 / CIFAR-10 | complete | 95.81 ± 0.25 / 0.157 ± 0.007 | **96.25 ± 0.34 / 0.122 ± 0.004** |
| | exp | 95.67 ± 0.11 / 0.162 ± 0.008 | **96.27 ± 0.18 / 0.118 ± 0.007** |
| | ring | 95.86 ± 0.31 / 0.161 ± 0.003 | **96.19 ± 0.02 / 0.119 ± 0.003** |

Table 7: Results with 16 workers after tuning the start epoch.

| Model / Dataset | Topology | Algorithm | |
|---|---|---|---|
| | | DSGD | DSGD-AC |
| WRN28-10 / CIFAR-100 | complete | 79.90 ± 0.51 / 1.046 ± 0.045 | **82.01 ± 0.21 / 0.744 ± 0.007** |
| | exp | 80.38 ± 0.28 / 0.980 ± 0.056 | **82.36 ± 0.45 / 0.732 ± 0.007** |
| | ring | 80.52 ± 0.45 / 0.948 ± 0.042 | **81.64 ± 0.06 / 0.843 ± 0.016** |
| WRN28-10 / CIFAR-10 | complete | 95.87 ± 0.31 / 0.200 ± 0.013 | **96.43 ± 0.24 / 0.135 ± 0.001** |
| | exp | 95.88 ± 0.09 / 0.194 ± 0.008 | **96.43 ± 0.05 / 0.130 ± 0.003** |
| | ring | 96.16 ± 0.10 / 0.180 ± 0.002 | **96.23 ± 0.18 / 0.134 ± 0.006** |
| WRN16-8 / CIFAR-10 | complete | 95.77 ± 0.16 / 0.172 ± 0.002 | **96.01 ± 0.29 / 0.120 ± 0.004** |
| | exp | 95.76 ± 0.09 / 0.159 ± 0.011 | **96.05 ± 0.10 / 0.131 ± 0.006** |
| | ring | 95.65 ± 0.30 / 0.163 ± 0.012 | **96.10 ± 0.38 / 0.130 ± 0.006** |

Table 8: Results with 32 workers after tuning the start epoch.

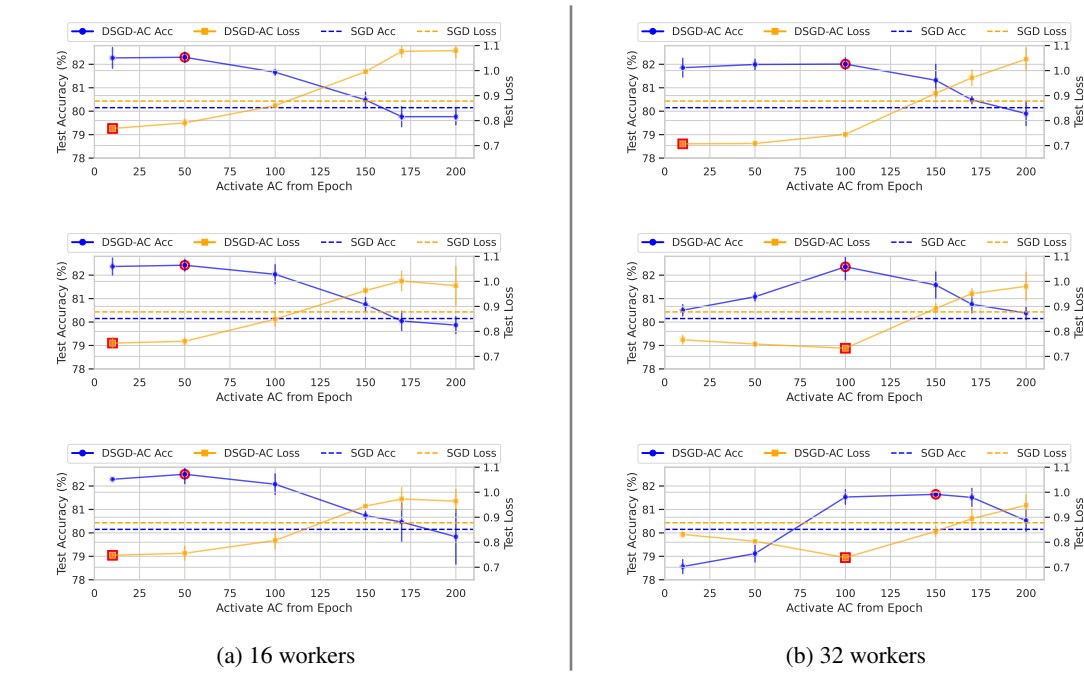

(a) 16 workers        (b) 32 workers

Figure 9: WRN28-10 on CIFAR100. Figures from top to bottom correspond to complete, exponential graph, and one-peer ring, respectively. The best test accuracy and the best test loss are highlighted by red marks.

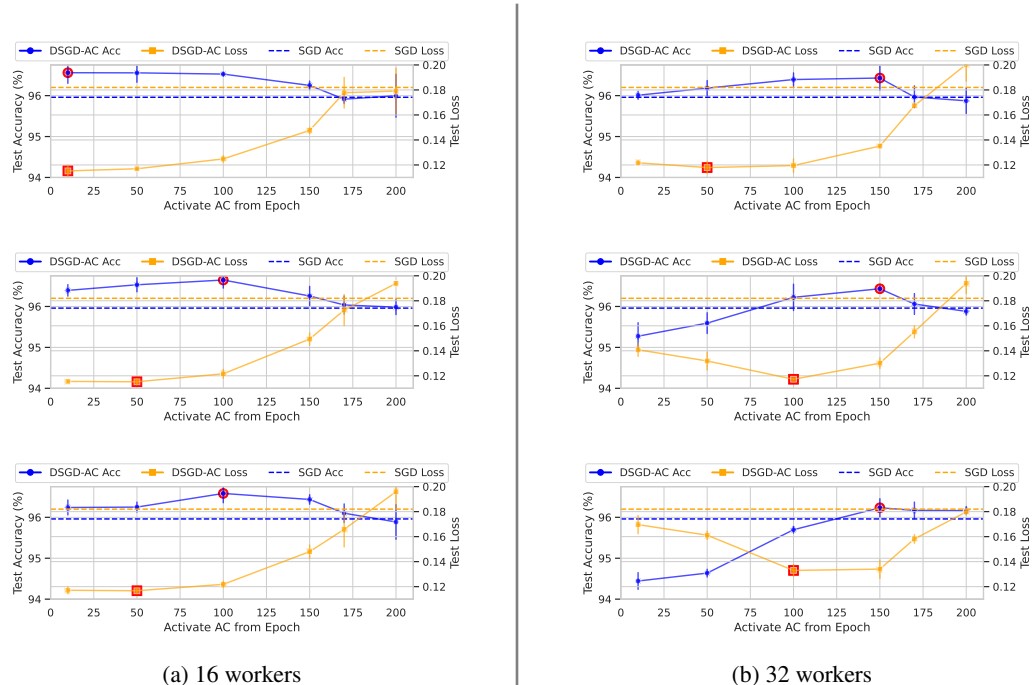

(a) 16 workers        (b) 32 workers

Figure 10: WRN28-10 on CIFAR10. Figures from top to bottom correspond to complete, exponential graph, and one-peer ring, respectively. The best test accuracy and the best test loss are highlighted by red marks.

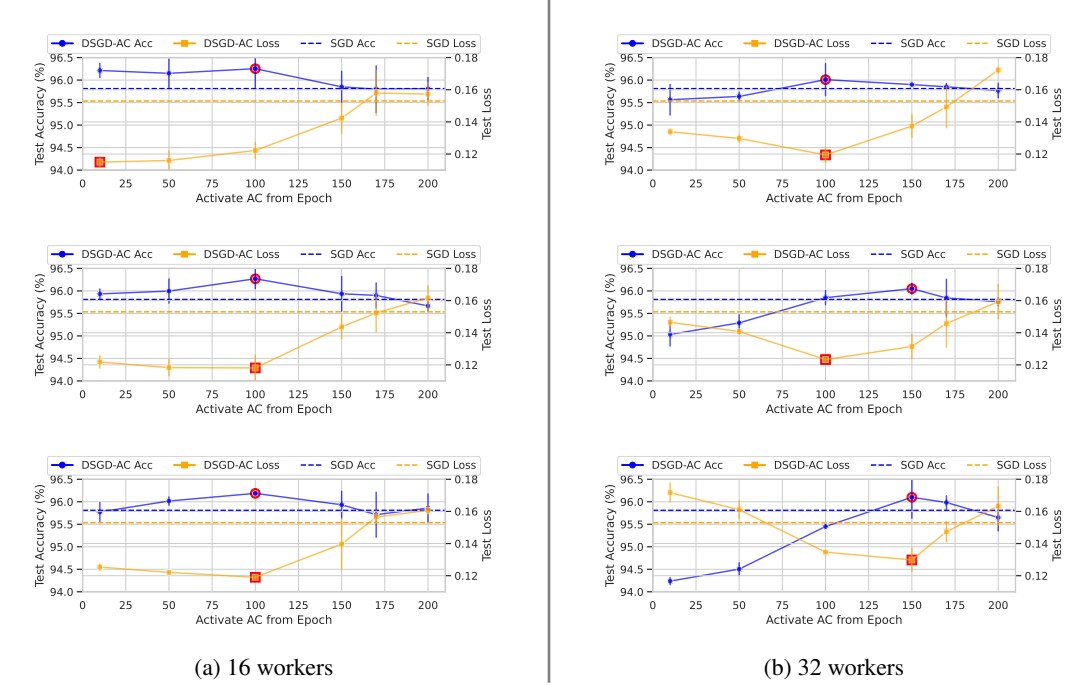

(a) 16 workers        (b) 32 workers

Figure 11: WRN16-8 on CIFAR10. Figures from top to bottom correspond to complete, exponential graph, and one-peer ring, respectively. The best test accuracy and the best test loss are highlighted by red marks.

### A.5.4 Sensitivity analysis of the hyperparameter in DSGD-AC

In all experiments, we use $p = 3$ for DSGD-AC, which is based on experiment tuning. The test results with $p = \{0, 1, 2, 3, 4, 5\}$ are presented in Figure 12 and Table 9. The tracked average norm of consensus errors with varying $p$ is shown in Figure 13.

Note that DSGD-AC with $p = 0$ is equivalent to DSGD. The results demonstrate the effectiveness of introducing $p$ and DSGD-AC, and $p = 3$ brings the best performance.

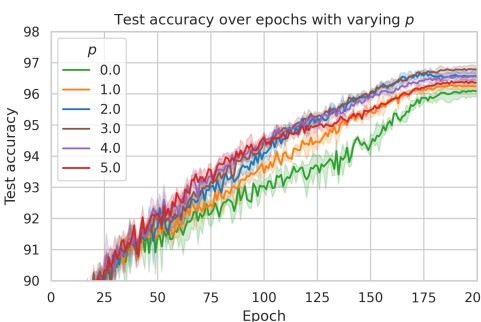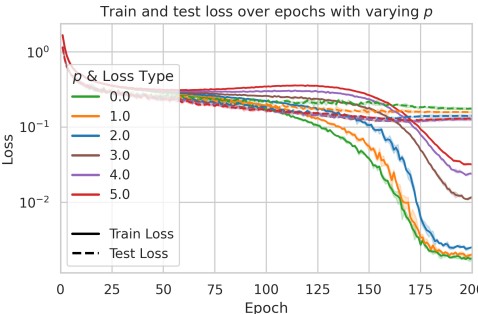

Figure 12: DSGD(-AC) on WRN28-10 on CIFAR-10 with varying $p$. **Left**: Test accuracy on test set. For decentralized training, the accuracy is evaluated on the global average model. **Right**: Training and test losses.

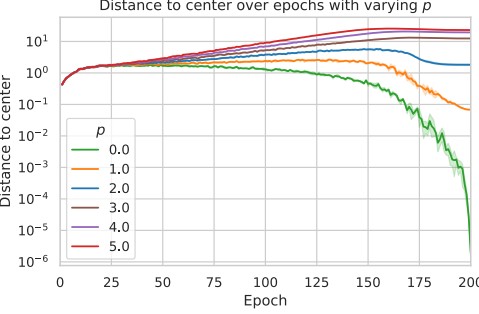

Figure 13: Average norm of consensus errors over epochs with varying $p$.

| $p$ | Test Accuracy (%) ↑ | Train Loss ↓ | Test Loss ↓ |
|---|---|---|---|
| 0 | $96.07 \pm 0.13$ | $\mathbf{0.002} \pm 0.000$ | $0.176 \pm 0.005$ |
| 1 | $96.26 \pm 0.14$ | $\mathbf{0.002} \pm 0.000$ | $0.159 \pm 0.003$ |
| 2 | $\underline{96.58} \pm 0.18$ | $0.003 \pm 0.000$ | $0.141 \pm 0.006$ |
| 3 | $\mathbf{96.77} \pm 0.11$ | $0.012 \pm 0.000$ | $\underline{0.128} \pm 0.003$ |
| 4 | $\underline{96.53} \pm 0.13$ | $0.024 \pm 0.001$ | $\mathbf{0.127} \pm 0.004$ |
| 5 | $96.37 \pm 0.04$ | $0.032 \pm 0.001$ | $0.130 \pm 0.002$ |

Table 9: Sensitivity analysis of parameter $p$ in the WRN28-10 on CIFAR10 experiment. The best value is **bold**, and the second best is underlined.

### A.6 DISCUSSION

**Future improvement directions**    The practicality in the adaptive consensus mechanism motivates the following future directions:

- Compression for communication–alignment tradeoffs. While communication compression in decentralized training has been widely studied (Koloskova et al., 2019; Vogels et al., 2020; Huang & Pu, 2024), most methods aim to approximate centralized training. DSGD-AC suggests a different view: small $\gamma^{(t)}$ and the alignment in Proposition 2 may benefit generalization. This opens the possibility of designing compressors that (i) spend more communication budget early in training when alignment forms, or (ii) implicitly maintain updates along high-curvature directions to further strengthen the alignment of disagreement with the dominant Hessian subspace.

- Decentralized mixing for better alignment. Current decentralized mixing relies on simple weighted averaging. Under DSGD-AC, one may interpret the disagreement as a curvature-related perturbation around the global model. This motivates exploring new mixing rules that selectively damp low-curvature disagreement while keeping high-curvature components active, thereby enhancing the "curvature tilt" observed in the algorithm. Such rules would be complementary to the compressors described above.

- Model fusion. Model fusion (Singh & Jaggi, 2020; Imfeld et al., 2023) combines models trained along different trajectories. For standard DSGD, their impact is limited because consensus errors quickly vanish, and the matching among parameters from local models is trivial. In DSGD-AC, however, the disagreement remains non-negligible, making model fusion a potential alternative to simple averaging, possibly improving performance.

**AC combined with adaptive optimizers**    In adaptive optimizers like Adam, the update is scaled by the inverse of the moving average of the componentwise square of the gradients. The scaling in each gradient coordinate eliminates the anisotropic structure in gradient noise (Zhou et al., 2010), which conflicts with the purpose of the AC mechanism which instead tends to enhance the structure. Since consensus errors are the accumulated updates after the scaling, the analysis in this paper may not directly work on the case that directly combines AC with adaptive optimizers. It can be an interesting direction for future work to find better ways for AC to co-exist with adaptive optimizers, possibly by recovering/extracting noise structure from the consensus errors. For example, designing the AC with variants like Adam-mini (Zhang et al., 2024) can be a practical idea for efficiently recovering the noise structure in the consensus errors.

### A.7 USE OF LARGE LANGUAGE MODELS

During the development of the paper, we used LLMs to polish the text without changing its original meaning.

