# OpenReview forum: "Controlled disagreement improves generalization in decentralized training"
_ICLR.cc/2026/Conference — Submitted to ICLR 2026_

### Official Review · Reviewer_vujS · 2025-10-29

**Soundness:** 2
**Presentation:** 3
**Contribution:** 3
**Rating:** 4
**Confidence:** 3

**Summary:**

This paper discovers and proves that the consensus errors in decentralized training align with the dominant Hessian subspace, acting as structured perturbations that guide optimization toward flatter minima. Building upon this insight, the authors propose DSGD-AC to maintain non-vanishing consensus errors throughout training. Experimental results show that this method achieves significantly improved generalization performance across multiple tasks.

**Strengths:**

1.	The paper provides an insightful interpretation of consensus errors in decentralized learning. Instead of considering these errors as harmful noise, the authors theoretically reveal their natural alignment with the dominant Hessian subspace and show that they implicitly serve the role of SAM-like perturbations.
2.	Experimental results show that DSGD-AC surpasses centralized SGD on multiple benchmarks.

**Weaknesses:**

1.	All experiments are conducted on the one-peer ring topology. Since consensus error behavior and spectral properties can vary substantially across different communication graphs, experiment results on other topologies (e.g., exponential graphs, fully-connected, random expander) would provide stronger evidence for the robustness of DSGD-AC.
2.	While Proposition 2 establishes a relationship between the step size $\alpha^{(t)}$, the Hessian spectrum, and the communication matrix $W$, this theoretical connection is not reflected in the adaptive factor $\gamma^{(t)}$ used in the algorithm. As a result, the paper does not yet exploit the network topology in its adaptive consensus mechanism.
3.	The design of the algorithm, especially the adaptive consensus factor $\gamma^{(t)}$ lacks the conceptual motivation and novelty.
4.	The paper lacks theoretical generalization bounds for DSGD-AC. The existing propositions mainly demonstrate that DSGD-AC maintains non-vanishing consensus errors, rather than providing guarantees on generalization performance.

**Questions:**

1.	On Page 13, Line 682, $V_k$ is treated as a time-invariant quantity, despite the fact that $\tilde{z}_k^{(t)}$ depends on the iteration $t$. It could be helpful if you could explain this.
2.	In the DAdam-AC experiments, the paper suggests that a more tailored adaptive consensus mechanism could further improve performance when combined with Adam. It would be better if you could elaborate on possible design directions.
3.	Proposition 2 provides a stability condition on the step size $\alpha^{(t)}$. It would be helpful if you could clarify how this condition is guaranteed to hold in the experiments.

---

> ### Author Response · Authors · 2025-11-20
> **Response to Reviewer vujS**
>
> We thank the reviewer for the thoughtful comments on our interpretation and experiments. We have updated the revision to address the concerns (please see "General Response" for details). Below, we respond to the concerns point by point.
>
> **W1 (Limited experiment results)**: We have updated sweeps on more workers and more topologies, and the experiment results confirm the practicality of the adaptive consensus. Please refer to the general response and our new revision (Appendix A.5.2 and A.5.3) for details.
>
> **W2 (Interpretation of Prop. 2)**: We have added Remark 1 in the new revision for a better interpretation of the stability condition in Proposition 2 and a better clarification of the theoretical benefit of DSGD-AC in enhancing the Hessian alignment of the consensus errors.
>
> **W3 (Conceptual motivation and novelty)**: We understand the reviewer’s impression that adding a time-dependent scaling factor may appear simple. However, the simplicity of the mechanism is precisely what makes the contribution practical and appealing: DSGD-AC changes only the scaling of a consensus term that is already present in decentralized training. In contrast, explicit sharpness-aware methods such as SAM [2] typically require an additional gradient evaluation per step, which increases implementation complexity and computational cost.
>
> More importantly, the key novelty lies not in the algebraic form of the update but in the theoretical and conceptual insight: by showing that consensus errors systematically align with dominant Hessian directions, we reveal that decentralization introduces a naturally structured perturbation that can be leveraged for regularization. DSGD-AC is the first method to explicitly control this effect rather than treat consensus errors as undesirable noise.
>
> **W4 (Lack of theoretical generalization bounds)**: We thank the reviewer for pointing out the lack of a formal generalization bound. We agree that such a bound would be valuable. However, deriving a meaningful generalization bound for decentralized training with non-vanishing consensus errors is highly nontrivial, because existing frameworks (e.g., stability-based or PAC-Bayesian analyses) typically assume either vanishing consensus error or centralized updates. In our setting, the maintained disagreement introduces a structured and time-varying perturbation that does not fit standard assumptions. We therefore focused our theoretical efforts on characterizing the geometry of the optimization trajectory.
>
> The newly added Appendix A.3 now provides a deterministic decomposition of the objective effectively optimized by DSGD-AC: locally, it minimizes the central loss plus a Hessian-weighted disagreement envelope whose weights are strictly increasing in curvature and more than linear in the Hessian eigenvalues (Corollary 1). We believe that this structure can be combined with existing stability analyses for DSGD (e.g., [1]) to derive formal generalization bounds, but we defer this to future work and now mention this explicitly as a limitation and a promising extension.
>
> **Q1 (Time-invariant $V_k$)**: We further clarified the assumption made in Proposition 1 and Appendix A.1 in the new revision. The analysis is carried out on short windows where ($\alpha^{(t)}$) and ($\gamma^{(t)}$) can be treated as approximately constant and the first two moments of the projected gradient noise are **slowly varying**, as is standard in steady-state analyses of SGD near a local minimum.
>
> **Q2 (DAdam-AC)**: Please refer to Appendix A.6 for a more detailed discussion about both the future development of AC and the potential problems when simply cooperating AC with adaptive optimizers like Adam.
>
> **Q3 (Stability condition)**: We agree that the design of the time-varying scaler $\gamma^{(t)}$ is not fully dedicated to fulfilling the condition in Proposition 2. Given the intuition from the sharpness-aware minimization methods, the consensus errors should ideally concentrate on the Hessian subspace spanned by eigenvectors corresponding to the top eigenvalues. Without an estimation (possibly costly) of the eigenvalues, it is difficult to enforce the stability condition during the training. In the newly added Remark 1, we clarified the interpretation of the condition and the theoretical benefit of the AC mechanism. The main idea is that the consensus errors tend to concentrate on the high-curvature directions more significantly compared to random noises, which enables "curvature tilted" in DSGD-AC with almost no computational overhead.
>
> [1] Le Bars, Batiste, et al. "Improved stability and generalization guarantees of the decentralized SGD algorithm." *arXiv preprint arXiv:2306.02939 (2023)*.
>
> [2] Foret, Pierre, et al. "Sharpness-aware minimization for efficiently improving generalization." *International Conference on Learning Representations* (2021).

---

### Official Review · Reviewer_yV2E · 2025-10-31

**Soundness:** 3
**Presentation:** 3
**Contribution:** 2
**Rating:** 4
**Confidence:** 4

**Summary:**

This paper's primary contribution is a novel perspective on the consensus step in decentralized SGD. The authors interpret the gossip-based parameter exchange as an implicit optimization on a quadratic consensus loss. Building on this, they argue that the *weight* of this implicit loss, which in standard DSGD is coupled with the inverse of the learning rate, should be controlled independently.

They introduce Decentralized SGD with Adaptive Consensus (DSGD-AC), which adds a time-dependent scaling factor $\gamma^{(t)}$ to the consensus term. The main goal of this modification is to prevent the consensus errors from vanishing as the learning rate decays, thereby preserving what prior work has identified as a "free sharpness regularization" inherent in the decentralized method. The authors provide theoretical support, arguing that these controlled, non-vanishing errors are not random noise but are structurally aligned with the dominant Hessian subspace, guiding the optimization toward flatter minima.

**Strengths:**

* The core observation is novel. Building on the work of Zhu et al. (2023), the idea that the implicit quadratic loss from the gossip phase should have its own adaptive scaling factor (decoupled from the main learning rate) is an insightful contribution.
* The paper provides a theoretical justification for the algorithm's design. Proposition 1 formally demonstrates that DSGD-AC, with the proposed scaling, can maintain non-vanishing consensus errors, which is the mechanism intended to preserve the free sharpness regularization.

**Weaknesses:**

* The paper's primary weakness lies in its experimental validation. The experiments are restricted to a very specific and limited setting: 8 nodes in a one-peer ring topology. This setup does not provide sufficient evidence for the method's effectiveness or scalability in more general or larger-scale scenarios (e.g., more nodes, different graph topologies).
* The performance improvement on the machine translation task is very small. In Table 2, the BLEU score for DAdam-AC is only marginally better than that of the standard Adam baseline.

**Questions:**

1.  Given that all experiments are on an 8-node ring, it is unclear how DSGD-AC would perform in other settings. Could the authors provide results or at least discuss the expected performance on larger systems (e.g., 16 or 32 nodes) or with different communication topologies (e.g., a torus or a more sparse random graph)?
2.  The proof for Proposition 1 relies on the assumption of a "quasi-stationary regime where the expectations and variances of the columns of $\tilde{G}^{(t)}$ remain constant." Could the authors elaborate on this assumption? Specifically, why is it a reasonable assumption in this context, how strong is it, and under what conditions during training might it fail to hold?

---

> ### Author Response · Authors · 2025-11-20
> **Response to Reviewer yV2E**
>
> We thank the reviewer for the careful review and constructive feedback. We have updated the revision to address the concerns (please see "General Response" for details), and we reply to each point below.
>
> **W1 & Q1 (weak experiment results)**: We have updated sweeps on more workers and more topologies, and the experiment results confirm the practicality of the adaptive consensus. Please refer to the general response and our new revision (Appendix A.5.2 and A.5.3) for details.
>
> **W2 (small improvement with DAdam-AC)**: With some more hyperparameter tuning (with $p=2$), DAdam-AC can achieve both better test loss and BLEU score (the primary metric for translation quality evaluation). We agree that the improvement for DAdam-AC is still modest compared with the case in DSGD-AC. We added a detailed discussion (Appendix A.6) of why combining AC with Adam is theoretically more subtle, and how optimizer-aware AC designs and Adam variants (e.g., Adam-mini) could further leverage the structured noise induced by consensus errors.
>
> **Q2 (Quasi-stationary regime assumption in Prop. 1)**: We further clarified this assumption in the statement of Proposition 1 and in Appendix A.1 in the new revision. The analysis is carried out on short windows where ($\alpha^{(t)}$) and ($\gamma^{(t)}$) can be treated as approximately constant and the first two moments of the projected gradient noise are slowly varying, as is standard in steady-state analyses of SGD near a local minimum. Under bounded-moment and spectral assumptions, this yields the scaling $r_t^2 = \Theta((\alpha^{(t)})^2/\gamma^{(t)})$. Empirically, we observe that the disagreement dynamics are indeed smoother in the later stages of training (Fig. 1), which is the regime our theory targets.

---

### Official Review · Reviewer_6ahb · 2025-10-31

**Soundness:** 2
**Presentation:** 3
**Contribution:** 3
**Rating:** 4
**Confidence:** 4

**Summary:**

This paper first empirically shows a positive correlation between consensus error and generalisability, which suggests that maintaining non-diminishing consensus error can boost generalisability. Based on this motivation, the authors design a new algorithm that has non-diminishing consensus error. They further theoretically show that this consensus error "align with dominant subspace of Hessian," which was shown linked to good generalisability. The paper also conducted experiments.

**Strengths:**

The method is interesting and well motivated, though it is a bit simple. The theory gives insightful message for designing algorithms though (still) a bit simple.

**Weaknesses:**

My first concern is that the theory has limited novelty and merit given an existing paper [Zhu et al 2023]. The previous paper has calculated the consensus error and proved the asymptotic equivalence between D-SGD and SAM. The new part in this paper is injecting a lambda to make the consensus error non-diminishing.

Further, the experimental results are not strong enough. The authors did not compare different topologies and hyperparameter alpha, two key tunable factors in this paper. The performance improvement is very marginal.

**Questions:**

It would be great if the authors can clarify the two weaknesses.

---

> ### Author Response · Authors · 2025-11-20
> **Response to Reviewer 6ahb**
>
> We appreciate the reviewer’s positive assessment of our motivation and methodological design. We have updated the revision to address the concerns (please see "General Response" for details). Below, we reply to the concerns about novelty and empirical evidence.
>
> **W1 (Limited novelty and merit vs [1])**: [1] showed an asymptotic equivalence between DSGD and average-direction SAM but treated consensus errors essentially as unstructured perturbations. In contrast, our paper:
> - **Sec. 3.1**: identifies the problem of diminishing consensus errors in exploiting the potential regularization effect.
> - **Prop.1 + Appendix A.1**: links the disagreement radius directly to $\gamma^{(t)}$ via $r_t^2 = \Theta((\alpha^{(t)})^2/\gamma^{(t)})$ and identifies $p\ge 2$ as necessary for non-vanishing consensus errors under decaying stepsizes.
> - **Prop.2 + Appendix A.2 + Remark 1**: analyzes the consensus-error dynamics in the joint Laplacian–Hessian eigenbasis and shows that disagreement energy concentrates in high-curvature modes, and proves it can be further enhanced by the adaptive consensus.
> - **Lemma 1, Prop. 3, Corollary 1 (Appendix A.3)**: derives a local loss envelope $F(\bar x^{(t)}) + \tfrac12 \operatorname{tr}(H\Sigma_t)$ and proves that the induced spectral weights are strictly increasing in the Hessian eigenvalues and more than linear in curvature.
>
> As for the simplicity of the proposed method, we agree that the algorithmic form of DSGD-AC is simple. However, this is a feature we view as a strength rather than a weakness. In the new revision, we further clarify why a simple time-varying scaler $\gamma^{(t)}$ suffices: Proposition 1 shows that $\gamma^{(t)}$ directly controls the disagreement radius, and Proposition 3 shows how the curvature-tilted envelope depends on the graph only through the Laplacian eigenvalues $\lambda_j(L)$ inside the weights $w_j(\lambda)$. This explains why a scalar factor already captures a rich interaction among curvature, noise, and topology. Moreover, DSGD-AC is one of the few decentralized algorithms that **intentionally maintains non-diminishing consensus errors** for better generalization performance.
>
> **W2**: We have updated sweeps on more workers and more topologies, and the experimental results demonstrate the practicality of the adaptive consensus. Please refer to the general response and our new revision (Appendix A.5.2 and A.5.3) for details.
>
> [1] Zhu, Tongtian, et al. "Decentralized SGD and average-direction SAM are asymptotically equivalent." *International Conference on Machine Learning.* PMLR, 2023.

---

### Official Review · Reviewer_jJrt · 2025-11-02

**Soundness:** 2
**Presentation:** 2
**Contribution:** 2
**Rating:** 2
**Confidence:** 4

**Summary:**

This paper proposes a novel algorithm called Decentralized SGD with Adaptive Consensus (DSGD-AC). The key idea is to introduce an adaptive scalar before the consensus term, which weakens the consensus process and intentionally maintains non-vanishing consensus errors. These errors are proven to be aligned with the dominant Hessian subspace. Empirically, the authors observe that, compared to DSGD and SGD, the proposed algorithm achieves higher test accuracy and converges to flatter minima on the CIFAR-10/CIFAR-100 image classification and WMT14 machine translation tasks.

**Strengths:**

The paper is easy to follow. The algorithm presented in Algorithm 1 is precise. The connection between the consensus error and the Hessian subspace is conceptually interesting.

**Weaknesses:**

1. **Overly strong assumptions**  The theoretical analysis relies on the assumption of *data homogeneity*—that all agents share identical local objectives $ f_i = f_j $.
   This assumption weakens the theoretical contribution.

2. **Lack of theoretical analysis for the adaptive scalar**   The adaptive scaling factor is taken as
   $\gamma^{(t)} = [\alpha^{(t)}/\alpha_{\max}]^p$.
  The theoretical analysis in this paper is not strong enough. The paper contains two propositions in the main section. Proposition 1 states that DSGD-AC's consensus error does not eventually drop to zero, unlike other DSGD algorithms. Proposition 2 states that the consensus error lies in the subspace spanned by the top eigenvectors of $H$. However, these two propositions do not convincingly demonstrate the necessity of using the adaptive scalar $\gamma^{(t)}$ in the main algorithm (Algorithm~1). I leave my detailed concerns about the propositions in the question section.

3. **Lack of significant experimental results**
   The experimental results are not sufficient. The authors conduct experiments only on small-scale problems, namely CIFAR-10 and WMT14. The results on WMT14 do not verify stronger performance of the adaptive consensus algorithm compared to ordinary centralized and decentralized methods. Some of my concerns about the experiments are discussed in the question section.

4. **Writing and notations** Some notations are confusing. In line 240, the authors use $L = U \Lambda U^\top$, while in line 310 they use the same symbol $H = U \Lambda U^\top$. Different symbols should be used for these different matrix decompositions.
In line 315, $\lambda_{\min}^W$ is used without definition.
In line 231, it should be $\Delta^{(t)} = X^{(t)}(I - \frac{1}{n}11^\top)$.

**Questions:**

1. In Proposition 2, the authors mention that by projecting the consensus error onto the $k$-th eigenvector $u_k$ (corresponding to the eigenvalue $\lambda_k$), the projected consensus error is linearly stable under the stability condition
$$
\alpha^{(t)} < \frac{2 + (\lambda_{\min}^{W} - 1)\gamma^{(t)}}{\lambda_k}.
$$
Here, $\lambda^{W}$ denotes the eigenvalue of the consensus matrix $W$. How does this proposition explain the advantage of DSGD-AC over ordinary DSGD? In ordinary DSGD, we have $\gamma^{(t)} \equiv 1$.

2. Suppose we set $W = I$, meaning that $n$ agents independently optimize the same objective function $F$ without communication. In this case, we have $\lambda_{\min}^{W} = 1$, and the right-hand side of the inequality becomes even larger. This suggests that separate training without any information exchange would be theoretically more stable than DSGD-AC. How should this be interpreted?

3. In Table 1, does ``SGD'' refer to centralized SGD? If so, why does DSGD achieve better testing accuracy and lower testing loss than SGD?

4. In Table 2, the result does not appear strong enough: DAdam-AC does not perform better than DAdam. Could the authors elaborate on this observation?

---

> ### Author Response · Authors · 2025-11-20
> **Response to Reviewer jJrt (Part 1/2)**
>
> We thank the reviewer for the careful reading and detailed feedback. We have updated the revision to address to concerns (please see "General Response" for details), and we reply to each weakness and question in turn below.
>
> **W1 (Overly strong assumptions)**: We agree that the assumption of identical local objectives ($f_i=F$) is strong for federated learning. In this work, however, we explicitly target cluster settings where a common dataset is available and is reshuffled and partitioned across workers each epoch; this yields i.i.d. local data distributions in expectation and is standard in decentralized training on GPU clusters (Sec. 2, "Practical remarks on data distributions and the distributed data sampler"). Our formal theory is therefore aligned with our experimental setting.
>
> At the same time, we agree that understanding heterogeneous data is important. While we do not develop a heterogeneity analysis in the paper, our linearized disagreement dynamics for each Laplacian–Hessian mode have an AR(1)-type form whose stationary variance we analyze in Proposition 1. Under standard bounded gradient (and mild Hessian) dissimilarity assumptions, heterogeneous objectives would add an extra driving term to this recursion, acting as additional noise/bias. Intuitively, this shifts the constants in the variance and induces a heterogeneity-driven floor on the disagreement radius, but does not change (1) the scaling ($r_t^2 = \Theta(\alpha(t)^2/\gamma^{(t)})$) that allows ($\gamma^{(t)}$) to control the radius, nor (2) the curvature-tilted envelope in Appendix A.3, which penalizes sharp directions more heavily. A careful, fully rigorous treatment of this heterogeneous case is beyond the scope of the current paper and is left as an interesting avenue for future work.
>
> **W2 (Lack of theoretical analysis for the adaptive scalar)**: We significantly strengthened the theory for ($\gamma^{(t)}$). Proposition 1 and Appendix A.1 now show that, in a quasi-stationary regime,
> $$
> r_t^2 = \Theta\Big(\frac{\alpha(t)^2}{\gamma(t)}\Big),
> $$
> so with ($\gamma^{(t)}$) bounded away from zero any decaying stepsize forces ($r_t^2 \to 0$). For $\gamma^{(t)}=g_0 (\alpha^{(t)})^p$, we derive the explicit regimes ($p<2$, $p=2$, $p>2$) and show that $p \ge 2$ is necessary to preserve a non-vanishing disagreement radius. This directly explains the role of the adaptive factor and why a constant $\gamma$ is insufficient when $\alpha(t)$ decays. In Algorithm 1, we instantiate this with ($p=3$), leading to a gently increasing radius late in training; Appendix A.5.4 confirms the predicted dependency on $p$ empirically.
>
> **W3 (Lack of significant experimental results)**: We addressed this by substantially expanding the experiments (App. A.5.2–A.5.4). On WRN28-10 / CIFAR-100 and CIFAR-10, and WRN16-8 / CIFAR-10, we now show results for 8, 16, and 32 workers and three different topologies (ring, exponential, complete). For 8 and 16 workers, DSGD-AC consistently improves over both DSGD and centralized SGD in test accuracy and loss on almost all settings; e.g., on CIFAR-100, 16 workers, complete graph, DSGD-AC improves test accuracy by about 2–3% and reduces test loss relative to both baselines. For 32 workers, we observe that activating AC from the start can hurt convergence when variance is very high, as expected. By tuning only the start epoch of AC (App. A.5.3), we restore and often improve performance across all topologies and worker counts: with a tuned start epoch, DSGD-AC outperforms both DSGD and centralized SGD in Tables 7–8.
>
> **W4 (writing and notations)**: We have addressed the weakness by fixing the writing and notation problems in the new revision.
>
> **Q1 (theoretical justification for AC)**: To address your concern, we have added Remark 1 in the new revision. It clarifies and highlights that a smaller $\gamma$ can enhance alignment with the dominant Hessian subspace compared to vanilla DSGD, as per Proposition 2.
>
> **Q2 (interpretation of $W=I$)**: We added Remark 2 in the new revision, which clarifies the regime where Proposition 2 is valid. With $W=I$, the disagreement radius can grow aggressively, and then the Taylor approximation made in Proposition 2 will no longer be valid.

---

> ### Author Response · Authors · 2025-11-20
> **Response to Reviewer jJrt (Part 2/2)**
>
> **Q3 (experiment results)**: Yes, SGD stands for centralized SGD. For the experiment results of DSGD and centralized SGD, we think it aligns with our propositions. Proposition 2 applies even when $\gamma=1$. With moderately poorly connected topology, even DSGD can, to some extent, enjoy the Hessian alignment property in the consensus errors before they diminish. In the experiments with 8 workers and one-peer ring topology, DSGD performs better than SGD on only two setups in the image classification task, and DSGD performs worse when scaling up the number of workers and with sparse topologies (see Appendix A.5.2 for details).
>
> Moreover, as discussed in Appendix A.4.1, an accurate evaluation of the deployed model benefits from recalibrating the batch-normalization statistics. A full calibration pass over the training set before validation substantially improves the test accuracy and loss reported for the deployed (averaged) model, especially early in training. This detail is often overlooked in decentralized CNN implementations, but now made explicit in our experimental protocol.
>
> **Q4**: On WMT14, DAdam-AC achieves the highest BLEU among the compared methods, and, after some tuning on $p$, DAdam-AC can also achieve the lowest test loss, though the gains are indeed smaller than on CIFAR. We now emphasize that BLEU is the main evaluation metric for translation, and we explain in Appendix A.6 why combining AC with Adam is more subtle: Adam’s per-coordinate adaptation can distort the anisotropic noise structure that AC is designed to exploit. We discuss concrete directions (e.g., coupling AC with Adam variants that better preserve curvature information) as future work.

---

> > ### Comment · Reviewer_jJrt · 2025-11-28
> > **Reply to rebuttal**
> >
> > Thank you for your detailed response and clarifications.
> >
> > In your revised version, the theorem is indeed strengthened. Proposition 1 provides a clearer intuition for the new term $\gamma^{(t)} = \left(\frac{\alpha^{(t)}}{\alpha_{\max}}\right)^p$ introduced in your algorithm. I also appreciate that you included experiments with varying $p$ in the appendix.
> >
> > However, I have several concerns regarding the theoretical aspects:
> >
> > **Conflicting Propositions:** While the authors have updated Proposition 1 and Proposition 2, I find that they are in conflict. Proposition 2 relies on a crucial assumption that $f(x_i^{(t)})\approx H(x_i^{(t)}-x^*)$, but Proposition 1 states that $r_t \to \infty$ when $p > 2$ (as proposed in the paper to maintain agent disagreement). This implies that $\|x_i^{(t)} - \bar{x}^{(t)}\| \to \infty$, which undermines the stability of the algorithm when $x_i^{(t)}$ is sufficiently close to $x^*$. In other words, choosing $p > 2$ will eventually cause $x_i^{(t)}$ to diverge significantly from $x^*$.
> >
> > **Strong Data Homogeneity Assumption:** The paper makes a very strong assumption regarding data homogeneity, which limits the generality of the theoretical results.
> >
> > Based on these points, I find the theoretical contribution of the paper to be weak.
> >
> > Regarding the experimental results:
> >
> > **Limited Baseline Comparison:** The experiments only compare DSGD-AC with vanilla DSGD. There is no comparison with state-of-the-art (SOTA) methods, nor is there any discussion of whether your approach can be combined with SOTA methods to further improve DSGD performance.
> >
> > Overall, the paper does not stand out in terms of theoretical or experimental contributions. Therefore, I will maintain my previous score.

---

> > > ### Author Response · Authors · 2025-11-28
> > > **Response to Reviewer jJrt (Part 1/2)**
> > >
> > > We thank the reviewer for taking the time to read our responses.
> > >
> > > 1. **"Conflicting" Propositions 1 and 2**
> > >
> > >    Propositions 1 and 2 are mathematically consistent and describe different aspects of the dynamics: Proposition 1 gives a scaling law for the magnitude of disagreement in a quasi-stationary regime, while Proposition 2 analyzes the structure of disagreement in a local neighborhood of a minimizer. We agree, however, that the current wording of Proposition 1 "$p>2 \Rightarrow r_t^2 \to \infty$" can be misleading when read outside the quasi-stationary context, and we will revise it in the camera-ready version to better reflect what is actually used in the paper.
> > >
> > >    More concretely, Proposition 1 is derived under an explicit quasi-stationary assumption and shows that, in that regime, $$r_t^2 =\Theta(\frac{(\alpha^{(t)})^2}{\gamma^{(t)}})$$
> > >    For the schedule $\gamma^{(t)} = g_0(\alpha^{(t)})^p$ this implies $$r_t^2=\Theta\big((\alpha^{(t)})^{2-p}\big)$$
> > >    so as the learning rate decreases, the radius shrinks for $p<2$, stays at a constant order for $p=2$, and grows as $\alpha^{(t)}$ decreases for $p>2$. The statement "$p > 2 \Rightarrow r_t^2 \rightarrow \infty$" in the current version is a shorthand for this scaling behavior inside the linear/quasi-stationary model; it is not meant as a claim that the full nonlinear algorithm will literally diverge when run indefinitely. In fact, if the disagreement radius became very large, the quasi-stationary and local-Hessian assumptions underlying Proposition 1 would themselves break down, so that the model should not be trusted all the way into that regime. For the purposes of this paper, what we actually need from Proposition 1 is the scaling ($r_t^2 = \Theta((\alpha^{(t)})^{2-p})$) and the qualitative fact that $p<2$ forces the disagreement radius to vanish asymptotically, whereas $p \ge 2$ allows it to remain at a nontrivial scale. We will adjust the wording of Proposition 1 accordingly, emphasizing this scaling and the lower bound $p \ge 2$ rather than the literal infinite-time limit.
> > >
> > >    Proposition 2, on the other hand, is explicitly a local result. There, we linearize the gradients around a strongly convex minimizer,
> > >    $$\nabla f_i(x_i^{(t)}) \approx H(x_i^{(t)} - x^*)$$
> > >
> > >     and analyze the mode-wise stability and structure of the consensus error under this approximation. This analysis is intended to hold on a time window where all iterates remain in a neighborhood of $x^*$ and the disagreement radius stays in a "reasonable" range; Remark 2 already notes that if the radius were pushed too large (for instance by taking ($\gamma^{(t)}$ too small for too long), the Taylor approximation would no longer be accurate and Proposition 2 would cease to apply. In other words, Propositions 1 and 2 are designed for different regimes: Proposition 1 tells us how the scale of disagreement behaves under a given ($\alpha^{(t)}$,$\gamma^{(t)}$) schedule in a quasi-stationary model, while Proposition 2 describes the shape of disagreement in a local convergent regime where the linearization is valid.
> > >
> > >    Importantly, in all our experiments, we operate exactly in this local, finite-horizon regime. We train for a fixed number of epochs (e.g., 200 on CIFAR-10/100) with a cosine schedule and a moderate choice ($p=3$). As shown in Fig. 1 (right) and in the sensitivity study in Fig. 13, the disagreement radius exhibits only a mild increase and remains bounded; the local models stay close to the global average rather than diverging. Moreover, Fig. 2 (right) shows that the loss along the line segment between the global average and each local model is well-approximated by a quadratic function over the range actually explored, supporting the validity of the gradient linearization used in Proposition 2. We will make these regime assumptions more explicit around Propositions 1-2 so that the connection to the experimental regime is clear even for readers who do not inspect the appendix in detail.
> > >
> > >    Note that this distinction between regimes is already present in the current manuscript. Remark 2 explicitly states that the Hessian-based analysis is intended only for a regime where the disagreement radius remains moderate and that the Taylor approximation breaks down if the disagreement becomes too large. In addition, the paragraph following Proposition 1 (lines 291–298 in the current draft) already discusses the limited validity of the local linear model and its connection to the choice of $p$. In the camera-ready version, we will revise these passages together with the wording of Proposition 1 to make this separation of regimes and its relation to the experiments more explicit.

---

> > > ### Author Response · Authors · 2025-11-28
> > > **Response to Reviewer jJrt (Part 2/2)**
> > >
> > > 2. **Strong data homogeneity assumption.**
> > >    We agree that assuming i.i.d. local objectives ($f_i \equiv F$) limits generality and will make this clearer in the paper. Nevertheless, we view this setting as both practically relevant and theoretically well-motivated:
> > > 	- **Relevance to distributed DNN training on GPU clusters.**
> > >
> > > 	  Our target setting is GPU-cluster training, where a single global dataset is shuffled and partitioned across workers—an environment very close to the i.i.d. assumption and consistent with recent decentralized training researches [1–5]. These works show that decentralized methods can improve throughput over centralized training, where communication is a major bottleneck [6]. Thus, analyzing generalization under i.i.d. partitions is directly relevant, and DSGD-AC is designed for this regime.
> > > 	- **A deliberate first step on the theory side.**
> > >
> > > 	  The i.i.d. case is the natural theoretical starting point, as it isolates consensus-error structure and Hessian alignment from the extra drift introduced by heterogeneous data. Prior analyses of consensus-induced regularization (e.g., [9,10]) also assume homogeneity. Our results extend this line by (i) characterizing how the disagreement radius scales with the adaptive consensus factor, and (ii) connecting consensus-error structure to the dominant Hessian subspace. Extending this spectral analysis to heterogeneous data, where additional bias terms must be controlled, is an interesting direction but beyond the *reasonable* scope of this work.
> > >
> > > 3. **Limited Baseline Comparison**:
> > >
> > >    We agree that baseline choice is important. In practice, centralized SGD is already a very strong reference point: decentralized methods generally aim to match its performance while improving throughput, and we are not aware of decentralized algorithms that consistently outperform a well-tuned centralized baseline on standard vision or translation tasks. We also are not aware of decentralized methods that share our design goal of intentionally maintaining non-vanishing consensus errors without adding computation or memory overhead beyond DSGD.
> > >
> > >    The paper already includes comparisons with several strong centralized methods. On CIFAR-10/100, we compare DSGD-AC not only with centralized SGD and standard DSGD, but also with sharpness-aware baselines such as average-direction SAM (AD-SAM) [7] and SAM [8] (Table 3, Appendix A.5.1). DSGD-AC achieves accuracy comparable to AD-SAM and somewhat below SAM; however, both AD-SAM and SAM are centralized and have worse runtime and communication efficiency than DSGD-AC, and SAM in particular incurs 2x computation cost per iteration. On WMT14, we similarly compare DAdam-AC against both centralized Adam and DAdam (Table 2). Our goal in this work is not to exhaustively benchmark all decentralized optimizers, but to introduce and study an adaptive consensus mechanism that is orthogonal to existing methods and could in principle be combined with them.
> > >
> > > [1] Lian, Xiangru, et al. "Can decentralized algorithms outperform centralized algorithms? a case study for decentralized parallel stochastic gradient descent." _Advances in neural information processing systems_ 30 (2017).
> > >
> > > [2] Assran, Mahmoud, et al. "Stochastic gradient push for distributed deep learning." _International Conference on Machine Learning_. PMLR, 2019.
> > >
> > > [3] Gan, Shaoduo, et al. "Bagua: scaling up distributed learning with system relaxations." arXiv preprint arXiv:2107.01499 (2021).
> > >
> > > [4] Ying, Bicheng, et al. "Bluefog: Make decentralized algorithms practical for optimization and deep learning." _arXiv preprint arXiv:2111.04287_ (2021).
> > >
> > > [5] Zesen Wang, Zhang Jiaojiao, Wu Xuyang, and Mikael Johansson. From promise to practice: realizing high-performance decentralized training. In *The Thirteenth International Conference on Learning Representations*. ICLR, 2025.
> > >
> > > [6] Huang, Tangsen, et al. "Optimizing energy consumption in centralized and distributed cloud architectures with a comparative study to increase stability and efficiency." _Energy and Buildings_ 333 (2025): 115454.
> > >
> > > [7] Bisla, Devansh, Jing Wang, and Anna Choromanska. "Low-pass filtering sgd for recovering flat optima in the deep learning optimization landscape." _International Conference on Artificial Intelligence and Statistics_. PMLR, 2022.
> > >
> > > [8] Foret, Pierre, et al. "Sharpness-aware minimization for efficiently improving generalization." _arXiv preprint arXiv:2010.01412_ (2020).
> > >
> > > [9] Zhu, Tongtian, Fengxiang He, Lan Zhang, Zhengyang Niu, Mingli Song, and Dacheng Tao. "Topology-aware Generalization of Decentralized SGD." *Proceedings of the 39th International Conference on Machine Learning (ICML)*, PMLR 162: 27479–27503, 2022.
> > >
> > > [10] Zhu, Tongtian, Fengxiang He, Kaixuan Chen, Mingli Song, and Dacheng Tao. "Decentralized SGD and Average-direction SAM are Asymptotically Equivalent." *Proceedings of the 40th International Conference on Machine Learning (ICML)*, PMLR 202: 43005–43036, 2023.

---

### Author Response · Authors · 2025-11-20
**General Response**

We sincerely thank all the reviewers for their thorough reviews and valuable feedback. We are glad to hear that our theoretical insights into consensus errors (Reviewers vujS and jJrt), the clarity and precision of our algorithmic presentation (Reviewer jJrt), the good empirical performance of DSGD-AC (Reviewer vujS), and the overall motivation and methodological design (Reviewer 6ahb) were positively received.

**We have addressed the comments and concerns from the reviewers in the new revision with all major changes marked in blue color.** The main changes are as follows:

1. **Stronger theory for the adaptive consensus factor:**
We revised Proposition 1 and Appendix A.1 to make the role of the adaptive factor ($\gamma^{(t)}$) explicit. In a quasi-stationary regime, we show that the disagreement radius
$$
r_t^2 := \mathbb{E}\Vert\Delta^{(t)}\Vert_F^2 = \Theta\Big(\frac{(\alpha^{(t)})^2}{\gamma^{(t)}}\Big),
$$
so, with a decaying stepsize $\alpha^{(t)}$, any constant $\gamma^{(t)}$ forces $r_t^2 \to 0$. For schedules $\gamma^{(t)} = g_0 (\alpha^{(t)})^p$, we derive
$$
p < 2 \Rightarrow r_t^2 \to 0,\quad
p > 2 \Rightarrow r_t^2 \to \infty,
$$
2. **Sharper structural interpretation of consensus errors:** Proposition 2 and Appendix A.2 reorganize the analysis in the joint Laplacian–Hessian eigenbasis and show that, under homogeneous objectives ($f_i = F$), small-curvature modes enter the stable regime earlier while high-curvature modes remain closer to instability and hence carry larger variance. We added Appendix A.3 showing that, near a local minimizer $x^*$,
$$
\frac{1}{N}\sum_i f_i(x_i^{(t)})
= F(\bar x^{(t)}) + \frac{1}{2} \operatorname{tr}(H \Sigma_t) + \text{higher-order terms},
$$
where $\Sigma_t$ is the disagreement covariance. Proposition 3 and Corollary 1 show that this quadratic envelope is a curvature-tilted penalty: its spectral weights are strictly increasing in the Hessian eigenvalues and larger eigenvalues receive disproportionately stronger penalties. This provides a principled interpretation of DSGD-AC as minimizing a central loss plus a Hessian-weighted regularizer induced by consensus errors.
3. **Clearer interpretation of stability condition in Proposition 2:** We added two remarks after Proposition 2 to clarify the theoretical benefits of the AC mechanism in enhancing the alignment property, and, to characterize the regime where the stability condition is valid and useful.
4. **Expanded and more diverse experiments:** Beyond the original CIFAR-10/100 and WMT14 experiments, we now:
   - **Vary the number of workers** ($n \in \{8, 16, 32\}$) and **topology** (one-peer ring, exponential graph, complete graph) on WRN28-10 / CIFAR-100, WRN28-10 / CIFAR-10, and WRN16-8 / CIFAR-10. Across 8 and 16 workers, DSGD-AC consistently improves test accuracy and loss over both DSGD and centralized SGD on almost all settings (with one mild exception, which we discuss), and often outperforms centralized SGD significantly (e.g., +2–3% on CIFAR-100).
   - Study the **start epoch** at which adaptive consensus is activated. For 16 and 32 workers we show that enabling AC too early can hurt convergence when variance is large, but tuning a single hyperparameter (the start epoch) restores and often enhances performance: with tuned start epoch, DSGD-AC outperforms both DSGD and centralized SGD across all three topologies and worker counts (Tables 7–8).

5. **Improved clarity and discussion:** We fixed the over-used eigenvector notation, added missing definitions, and corrected the typos pointed out by the reviewers. We also added a discussion section (App. A.6) that elaborates on future directions about alignment-aware consensus, communication-alignment tradeoffs, model fusion, and better integration with adaptive optimizers.

We believe these changes address the main concerns about the soundness, novelty, and empirical support of DSGD-AC.

Below, we provide detailed, point-by-point responses to each reviewer’s weaknesses and questions. We hope these revisions and clarifications resolve the main concerns, and we would be grateful for any additional feedback.

---

### Meta-Review · Area_Chair_qmaJ · 2026-01-06

**Summary:**

1. [jJrt, 6ahb, yV2E, vujS] The experimental results are not sufficient. The authors conduct experiments only on small-scale problems, namely CIFAR-10, CIFAR-100, and WMT14. The results on WMT14 do not verify stronger performance of the adaptive consensus algorithm compared to ordinary centralized and decentralized methods. The experiments are restricted to a very specific and limited setting: 8 nodes in a one-peer ring topology. This setup does not provide sufficient evidence for the method's effectiveness or scalability in more general or larger-scale scenarios (e.g., more nodes, different graph topologies).
2. [jJrt] The analysis relies on the overly strong assumption that all agents share identical local objectives.
3. [jJrt] The theoretical analysis of the adaptive scaling factor is insufficient. The paper contains two propositions in the main section. Proposition 1 states that DSGD-AC's consensus error does not eventually drop to zero, unlike other DSGD algorithms. Proposition 2 states that the consensus error lies in the subspace spanned by the top eigenvectors of $H$. However, these two propositions do not convincingly demonstrate the necessity of using the adaptive scalar in Algorithm 1.
4. [6ahb] My first concern is that the theory has limited novelty and merit given an existing paper [Zhu et al 2023]. The previous paper has calculated the consensus error and proved the asymptotic equivalence between D-SGD and SAM. The new part in this paper is injecting a lambda to make the consensus error non-diminishing.
5. [vujS] The design of the algorithm, especially the adaptive consensus factor $\gamma^{(t)}$ lacks the conceptual motivation and novelty. While Proposition 2 establishes a relationship between the step size $\alpha^{(t)}$, the Hessian spectrum, and the communication matrix $W$, this theoretical connection is not reflected in the adaptive factor $\gamma^{(t)}$ used in the algorithm. As a result, the paper does not yet exploit the network topology in its adaptive consensus mechanism.
6. [jJrt] In ordinary DSGD, we have $\gamma^(t) \equiv 1$. Suppose we set $W = I$, meaning that agents independently optimize the same objective function $F$ without communication. In this case, we have $\lambda^{W}_{min} = 1$, and the right-hand side of the inequality becomes even larger. This suggests that separate training without any information exchange would be theoretically more stable than DSGD-AC. How should this be interpreted?
7. [vujS] The paper lacks theoretical generalization bounds for DSGD-AC. The existing propositions mainly demonstrate that DSGD-AC maintains non-vanishing consensus errors, rather than providing guarantees on generalization performance.
8. [yV2E] The proof for Proposition 1 relies on the assumption of a "quasi-stationary regime where the expectations and variances of the columns of $\tilde{G}^{(t)}$ remain constant." Could the authors elaborate on this assumption? Specifically, why is it a reasonable assumption in this context, how strong is it, and under what conditions during training might it fail to hold?
9. [vujS] On Page 13, Line 682, $V_{k}$ is treated as a time-invariant quantity, despite the fact that $\tilde{z}_{k}^{(t)}$ depends on the iteration $t$. It could be helpful if you could explain this.
10. [jJrt] In Proposition 2, the authors mention that by projecting the consensus error onto the $k$-th eigenvector (corresponding to the eigenvalue $\lambda_{k}$), the projected consensus error is linearly stable under the stability condition. How does this proposition explain the advantage of DSGD-AC over ordinary DSGD?
11. [vujS] Proposition 2 provides a stability condition on the step size $\alpha^{(t)}$. It would be helpful if you could clarify how this condition is guaranteed to hold in the experiments.
12. [jJrt] Writing and notations Some notations are confusing. In line 240, the authors use $L = U \Lambda U^{T}$, while in line 310 they use the same symbol $H = U \Lambda U^{T}$. Different symbols should be used for these different matrix decompositions. In line 315, $\lambda^{W}_{min}$ is used without definition. In line 231, it should be $\Delta^{(t)} = X^{(t)}(I - \frac{1}{n}11^{T})$.
13. [vujS] In the DAdam-AC experiments, the paper suggests that a more tailored adaptive consensus mechanism could further improve performance when combined with Adam. It would be better if you could elaborate on possible design directions.

**Reviewer Concerns:**

1. This concern is partially addressed in the rebuttal and revision. The authors (1) added experimental results on CIFAR-10 and CIFAR-100 for a wider range of topologies and workers, (2) improved WMT14 results by tuning $p$, (3) showed that for a larger number of workers (16 or 32) it is best to enable the adaptive consensus mechanism after several epochs of training without it, and (4) added comparisons to AD-SAM and SAM. However, the limited range of tasks considered in the empirical evaluation is still a serious weakness of the paper.
2. This concern is fully addressed in the rebuttal and revision: the authors clarify that the focus of the paper is on training scenarios in which all agents have access to the full training dataset, so the assumption is not restrictive in this case. While Reviewer jJrt complained in a response to the rebuttal that the strong data homogeneity assumption is too restrictive, the AC thinks that relaxing that assumption is beyond the scope of the current paper because there are plenty of practical applications of DSGD where all agents have access to the full training dataset.
3. This concern is fully addressed in the rebuttal and revision: the authors derive a bound on the disagreement radius and illustrate how the choice of $p$ affects its dynamics during training.
4. This concern is fully addressed in the rebuttal and revision: the authors emphasize that while the previous literature treated consensus errors as unstructured, the current paper carefully analyzes the structure of the consensus errors and shows that they are beneficial because they drive iterates towards regions of the loss landscape having lower curvature, and also add a derivation showing how the choice of $p$ affects the dynamics of the disagreement radius during training.
5. This concern is fully addressed in the rebuttal and revision: the authors emphasize that the simplicity of the proposed DSGD-AC algorithm follows from their analysis of the properties of the consensus errors and demonstration that non-zero consensus error helps steer the optimization to flatter parts of the loss landscape. They also clarify their presentation of Proposition 2, adding Remark 1 in the new revision for a better interpretation of the stability condition in Proposition 2 and a better clarification of the theoretical benefit of DSGD-AC in enhancing the Hessian alignment of the consensus errors.
6. This concern is fully addressed in the rebuttal and revision: the authors add Remark 2 in the new revision, which clarifies the regime where Proposition 2 is valid. This regime excludes $W = I$.
7. This concern is partially addressed in the rebuttal and revision. The authors added Appendix A.3, which provides a deterministic decomposition of the objective effectively optimized by DSGD-AC and explain that they believe this loss envelope can be combined with existing stability analyses for DSGD to derive formal generalization bounds.
8. This concern is addressed in the revision and rebuttal. The authors explain that they clarified this assumption in the statement of Proposition 1 and in Appendix A.1 in the new revision. Specifically, they say that the analysis is carried out on short windows where $\alpha^{(t)}$ and $\gamma^{(t)}$ can be treated as approximately constant and the first two moments of the projected gradient noise are slowly varying, as is standard in steady-state analyses of SGD near a local minimum. Empirically, the authors observe that the disagreement dynamics are indeed smoother in the later stages of training (Fig. 1).
9. This concern is addressed in the revision and rebuttal. The authors explain that they clarified this assumption in the statement of Proposition 1 and in Appendix A.1 in the new revision. Specifically, they say that the analysis is carried out on short windows where $\alpha^{(t)}$ and $\gamma^{(t)}$ can be treated as approximately constant and the first two moments of the projected gradient noise are slowly varying, as is standard in steady-state analyses of SGD near a local minimum. Empirically, the authors observe that the disagreement dynamics are indeed smoother in the later stages of training (Fig. 1).
10. This concern is fully addressed in the revision and rebuttal. The authors added added Remark 1, which highlights that a smaller $\gamma$ enhances alignment with the dominant Hessian subspace compared to vanilla DSGD, as per Proposition 2.
11. This concern is partially addressed in the revision and rebuttal. The authors explain that their analysis shows that the consensus errors tend to concentrate on the high-curvature directions more significantly compared to random noises, which enables "curvature tilted" in DSGD-AC with almost no computational overhead.
12. This concern was fully addressed: the authors corrected their notation and the errors highlighted by the reviewer.
13. This concern was addressed in the rebuttal and revision: the authors added a discussion of future work on applying adaptive consensus to optimizers such as Adam.

**Reviewer Scores:**

**jJrt** - In the discussion that occurred before the cutoff, this reviewer stated that they were not going to change their score, and I do not believe that the authors' response to this statement would have changed the reviewer's mind.

**6ahb** - I do not believe that this reviewer would have increased their score, given their apparent opinion that the proposed algorithm follows directly from Zhu et al.

**yV2E** -I believe this reviewer would have increased their score because the authors capably addressed their concerns and the reviewer already stated that the core idea builds nicely on Zhu et al. and is an insightful contribution.

**vujS** - I believe that this reviewer would not have increased their score because of the missing generalization bounds.

---

### Decision · Program_Chairs · 2026-01-26

Reject